# Dissipation-driven integrable fermionic systems: from graded Yangians to exact nonequilibrium steady states

**Enej Ilievski**

Institute for Theoretical Physics Amsterdam and Delta Institute for Theoretical Physics, University of Amsterdam, Science Park 904, 1098 XH Amsterdam, The Netherlands

⋆ e.ilievski@uva.nl

## Abstract

Using the Lindblad master equation approach, we investigate the structure of steady-state solutions of open integrable quantum lattice models, driven far from equilibrium by incoherent particle reservoirs attached at the boundaries. We identify a class of boundary dissipation processes which permits to derive exact steady-state density matrices in the form of graded matrix-product operators. All the solutions factorize in terms of vacuum analogues of Baxter's Q-operators which are realized in terms of non-unitary representations of certain finite dimensional subalgebras of graded Yangians. We present a unifying framework which allows to solve fermionic models and naturally incorporates higher-rank symmetries. This enables to explain underlying algebraic content behind most of the previously-found solutions.

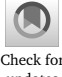
# 1  Introduction

Remarkable progress in experiments with cold atoms [1–7] has greatly impacted theoretical research in the area of quantum many-body dynamics [8–16]. Quantum systems which reside in the proximity of a quantum integrable point have received a great amount of attention. Non-ergodic character of these systems was revealed through anomalous relaxation and absence of conventional thermalization, and paved the way to study new paradigms in quantum statistical mechanics such as pre-thermalization [17–22] and equilibration towards generalized Gibbs ensembles [23–32]. In an idealized scenario which neglects integrability-breaking perturbations, integrable interacting systems were shown to permit a universal classification of local equilibria [33–35] based on a complete set of local conservation laws [36].

Equilibrium statistical ensemble however constitute a fairly small set of quantum many-body states and are outside of perturbative regime insufficient to capture physically interesting situations in which systems support particle and energy currents. An important step towards realizing genuine far-from-equilibrium regimes is to devise an efficient computational framework for accessing regimes of strongly-correlated quantum dynamics which often lie beyond the reach of traditional techniques. Switching from the Hamiltonian approach for closed systems to the *open system* perspective [37–39] offers a promising route to achieve this.

A quantum system is regarded as an open system when as a result of interactions with its surroundings experiences incoherent loss of information (quantum decoherence), making a system evolving according to an effective *non-unitary* evolution law. In a highly controlled environment however, an irretrievable loss of information due to quantum noise may sometimes even act as a resource [40–43]. Quantum noise is typically modelled either as a stochastic process, or alternatively via deterministic evolution laws in the form of quantum master equations, where the unitary dynamics is supplemented with additional non-Hamiltonian effective terms. The simplest master equations are Markovian [44,45] and thus entirely discard memory effects between a system and its environment.

Quantum master equations can be in many aspects perceived as quantum analogues of classical stochastic models [46]. The latter encompass a large class of systems which include asymmetric simple exclusion processes [47,48], reaction-diffusion processes [49–52], zero-range processes [53,54] and others (see e.g. [55] for a review). While classical stochastic equations have been a subject of intense research in the past few decades which has lead to many ex-

actly solvable examples [47, 48, 56–61], it is quite surprising that there exist merely a handful of recent theoretical studies of quantum master equations in the realm of low-dimensional many-particle systems [62–69].

Despite quantum many-body systems which undergo dissipation typically evolve to either trivial states, or highly entangled states of prohibitive complexity, there remarkably exist certain non-trivial examples of quantum dissipative Markovian dynamics where an intricate interplay between noise and coherent evolution results in stationary states which are analytically tractable. Integrability of the central model is of central importance here, as it makes it possible to identify Markovian particle reservoirs which, as explained in the manuscript, induce certain 'symmetry protected' nonequilibrium states. The main objective in this regard is to isolate scenarios in which the steady states of dissipative many-body dynamics are of low complexity and permit an exact analytic description in terms of matrix-product states. This programme has been initially pursued in classical exclusion processes [47, 48] and relatively recently applied to a quantum chain of non-interacting fermions [70, 71]. The same ideas have been shortly after expanded also to a few representative interacting exactly solvable many-body Hamiltonians, such as the Heisenberg spin chain [72–74] and the fermionic Hubbard model [75, 76], which led to various applications (see e.g. [72, 77–79]). For a historical perspective on the subject the reader is referred to the recent topical review [80].

Despite many promising advancements on the subject, it is rather unsatisfactory that the structure of these solutions still remains elusive and poorly understood. Indeed, no common framework which would explain the origin and meaning of integrable dissipative boundaries and offer a systematic way to extend the results to more general scenarios has been proposed to the date. In particular, all previous attempts to understand the internal structure of these exactly solvable instances based on 'first symmetry principles' and algebraic concepts of Yang–Baxter integrability have been mainly unsuccessful, although a few central insights have been made in [81, 82] which unveiled the Lax formulation and highlighted the importance of non-unitary representations of quantum groups. However, a comprehensive group-theoretic approach which would enable to construct a larger class of solutions from first principles remains unknown.

The primary goal of this work is to study formal aspects of integrable quantum chains driven far from equilibrium with aid of incoherent Markovian reservoirs attached at their ends. By continuing the bottom-up approach initiated in [81], we shall uncover the symmetry content behind some of the solutions obtained in the previous works, extend these results to models based on higher rank algebras and discuss the paradigm of integrable steady states from the standpoint of representation theory of quantum algebras. This work offers a unifying algebraic construction for an entire class of exact nonequilibrium states belonging to the so-called rational integrable quantum spin chains by making use of tools of quantum integrability theory. Specific instances which have been derived in the previous work with different techniques(cf. [73,74,81,82]) are thus naturally incorporated in a common framework. Moreover, by using graded vectors spaces and Lie superalgebras, we present how to accommodate for fermionic degrees of freedom [83, 84] and derive a new class of steady-state solutions for interacting integrable fermionic chains with $SU(n|m)$-symmetric Hamiltonians. Further quantitative analysis of the constructed solutions goes beyond the main scope of the present study and will be thus omitted. The task of computing correlation functions can however be carried out by using standard techniques based on matrix-product states, see e.g. [80].

One of the central insights of our approach is rooted in the universal factorization property of quantum Lax operators. This leads to the so-called 'partonic' Lax operators which can be regarded as the elementary constituents of Yang–Baxter integrable systems and are intrinsically related to the notion of Baxter's Q-operators [85–88], a widely used concept in the Bethe Ansatz diagonalization techniques. The observation that partonic Lax operators which realized

over non-unitary irreducible modules may be used as local building units of exact steady-state solutions to certain Lindbladian dynamics is however a curious unconventional feature which displays their proper nonequilibrium character.

**Outline.** The paper is organized as follows. In the preliminary section 2 we give a quick introduction to the Lindblad master equation and briefly review some basic concepts regarding graded vectors spaces. In section 3, we proceed by introducing an out-of-equilibrium protocol by coupling a quantum chain of interacting particles to incoherent particle reservoirs attached at its ends. We subsequently present the main algebraic structures which are afterwards used in the construction of the steady-state solutions. The notion of graded Yangians is defined in section 4, where finite dimensional subalgebras which are intimately related to the Baxter's Q-operators are identified. In section 5 we outline a unifying construction for a class of non-trivial current-carrying steady states, and present a few explicit examples of widely studied integrable spin and fermionic chains. In section 6 we provide some technical remarks on the notion of vacuum Q-operators, and conclude in section 7 by summarizing the main results and providing an outlook.

## 2 Preliminaries

### 2.1 Lindblad master equation

In the approach of open quantum systems [38, 39], the time-evolution of a density operator $\rho(t)$ of a central system (which presently represents a one-dimensional system of spins or interacting fermions) is governed by a completely positive and trace-preserving map $\mathscr{V}(t)$, reading compactly

$$\rho(t) = \mathscr{V}(t)\rho(0), \qquad \mathscr{V}(t) = \exp(t\mathscr{L}). \tag{1}$$

The Liouville propagator obeys the semi-group property $\mathscr{V}(t_1 + t_2) = \mathscr{V}(t_2)\mathscr{V}(t_1)$. Notice that, in contrast to the unitary propagator of the Hamiltonian evolution, the generator $\mathscr{V}(t)$ is not invertible. The generator $\mathscr{L}$ takes the *Lindblad form* [44, 45]

$$\mathscr{L} = \mathscr{L}_0 + \mathscr{D}, \tag{2}$$

where $\mathscr{L}_0 \rho \equiv -\mathrm{i}[H, \rho]$ is the ordinary Liouville–von Neumann unitary dynamics generated by the Hamiltonian $H$ of the central system, while $\mathscr{D}$ is the dissipator which fully encodes an *effective* description of the environment and admits a canonical resolution in terms of the Lindblad operators $A_k$,

$$\mathscr{D}\rho = \sum_k \left( \left[ A_k, \rho A_k^\dagger \right] + \left[ A_k \rho, A_k^\dagger \right] \right). \tag{3}$$

Here each Lindblad 'jump operator' $A_k$ acts as an independent incoherent process.[1] In our application, $A_k$ will be used to model incoming and outcoming particle flows through the boundaries of the quantum chain. A particular advantange of such a nonequilibrium protocol is to have a simple setup for obtaining exact or approximate results for genuine far-from-equilibrium states, reaching beyond the traditional linear response theory and quasi-stationary regimes described with the hydrodynamic approach [13, 92, 93].

In this work we shall exclusively restrict our considerations to the *steady states*. The latter correspond, by definition, to fixed points of the Liouville dynamics, $\rho_\infty = \lim_{t\to\infty} \rho(t)$. This

---

[1] Lindbladian flows can be alternatively understood in terms of 'quantum trajectories', i.e. an approach which uses a stochastic differential equation for an ensemble of pure quantum states evolving under an effective non-hermitian Hamiltonian [89–91].

means that a steady state is an operator $\rho_\infty$ from the *kernel* (null space) of the generator $\mathscr{L}$,

$$\mathscr{L}\rho_\infty = 0. \tag{4}$$

We will also encounter situations when $\dim \ker \mathscr{L} > 1$, which physically corresponds to degenerate steady states and leads to higher dimensional steady-state manifolds.

## 2.2 Graded vector spaces

In order to incorporate fermionic degrees of freedom in our description we shall make use of graded vector spaces. A local Hilbert space attached to a site in a quantum chain is denoted by $\mathbb{C}^{n|m}$, where integers $n$ and $m$ in the superscript signify the number of bosonic and fermionic states, respectively. Below we briefly recall a few basic notions of graded vectors space and refer the reader for a more detailed exposition to appendix A.

The two types of states are distinguished by the $\mathbb{Z}_2$-parity,

$$p: \quad \{1, 2, \ldots, n + m\} \to \{0, 1\}. \tag{5}$$

The mapping $p$ equips $\mathbb{C}^{n+m}$ with a $\mathbb{Z}_2$-grading: if $a$ belongs to a subset of bosonic (fermionic) indices we assign it a parity $p(a) = 0$ ($p(a) = 1$). Gradation is naturally lifted to vector spaces $\mathbb{C}^{n+m}$ and furthermore to the Lie algebra of linear operators acting on $\mathbb{C}^{n+m}$. Specifically, by adopting the distinguished grading in which $p(a) = 0$ for $a \in \{1, 2, \ldots n\}$ and $p(a) = 1$ for $a \in \{n + 1, \ldots m\}$, the space of $(n + m)$-dimensional matrices on $\mathbb{C}^{n+m}$ block-decomposes into the bosonic (even) subspace $\mathscr{V}_0$ and fermionic (odd) subspace $\mathscr{V}_1$. The two subspaces are typically referred to as the homogeneous components. The fundamental $\mathfrak{gl}(n|m)$ representation, denoted by $\mathscr{V}_\square^{n|m}$, is spanned by a basis of matrix units $E^{ab}$, $(E^{ab})_{ij} = \delta_{ai}\delta_{jb}$. The action of the Lie bracket adjusted to the grading is expressed as

$$\left[E^{ab}, E^{cd}\right] = \delta_{cb}E^{ad} - (-1)^{(a+b)(c+d)}\delta_{ad}E^{cb}. \tag{6}$$

Since exchanging two fermionic states results in a minus sign, the presence of fermionic states in a graded tensor product space non-trivially affects the multiplication rule. Namely, for a set of homogeneous elements[2] we have

$$(A \otimes B)(C \otimes D) = (-1)^{BC}(AC \otimes BD). \tag{7}$$

Tensor multiplication can be conveniently recast in the standard from by introducing the graded tensor product $\circledast$, defined in accordance with $(A \circledast B)(C \circledast D) = AC \circledast BD$. Further clarifications about the notation can be found in appendix A.

## 3 Exactly solvable nonequilibrium steady states

The algebraic construction of the solutions which is outlined below consists of two steps. The general strategy in some sense reminds of solving a Poisson's equation. Namely, the first step is to identify a space of solutions for the bulk part which only accounts for the unitary part of the generator $\mathscr{L}_0$. Note that the entire space of bulk solutions is determined purely from the kinematic constraints, i.e. it is determined solely from the quantum symmetry algebra of the spin chain, irrespective of the representation labels. The second step is to impose the dissipative boundary conditions which (when a solution exists) uniquely fixes the physical state at hand. More specifically, this step amounts to chose suitable boundary auxiliary states

---

[2] Homogeneous elements are linear operators on $(\mathbb{C}^{n|m})^{\otimes N}$ with a well-defined parity.

and subsequently solve a non-linear system of boundary constraints in the space of the free representation parameters. Such a separation of bulk and boundary processes is indeed a characteristic feature of all exactly solvable classical and quantum boundary-driven lattice models.

The aim of this section is to break down the entire procedure into elementary steps and systematically discuss all the necessary ingredients to carry out the algebraic construction for the class of steady-state solutions of integrable quantum chains. The more difficult problem of identifying and classifying the relevant class of subalgebras is postponed to the next section, before finally presenting a few explicit results in Section 5.

## 3.1 Graded Yang–Baxter relation

This work is focused on a particular class of integrable lattice models which involve both bosonic and fermionic states. These models can be systematically derived from the so-called rational solutions to the *graded Yang–Baxter relation*. On a two-particle space $\mathbb{C}^{n|m} \otimes \mathbb{C}^{n|m}$ the latter takes the following form

$$R^{n|m}(z_1 - z_2)\big(\mathbf{L}(z_1) \circledS 1\big)\big(1 \circledS \mathbf{L}(z_2)\big) = \big(1 \circledS \mathbf{L}(z_2)\big)\big(\mathbf{L}(z_1) \circledS 1\big)R^{n|m}(z_1 - z_2), \qquad (8)$$

where $z_{1,2}$ are two arbitrary complex numbers usually referred to as the spectral parameters. Here and subsequently we shall use the convention in which bold-faced symbols pertain to operators which act non-identically in the auxiliary space(s).

Let us first explain the main objects. The graded $R$-matrix $R^{n|m}(z)$ acts as an intertwiner on the two-fold space $\mathbb{C}^{n|m} \otimes \mathbb{C}^{n|m}$, i.e. expresses the equivalence of two distinct orderings of the tensor product of two $\mathbf{L}$-operators. Matrices $R^{n|m}(z)$ are simply related to the graded permutation matrices $P^{n|m}$,

$$R^{n|m}(z) = z + P^{n|m}, \qquad P^{n|m} = (-1)^b E^{ab} \circledS E^{ba} = (-1)^{ab} E^{ab} \otimes E^{ba}, \qquad (9)$$

where matrix units $E^{ab}$ form the standard basis of linear operators in the fundamental module $\mathcal{V}_{\square}^{n|m}$. The rational Yang–Baxter relation (8) can be formally understood as the *defining relation* of an infinite-dimensional associative algebra $\mathcal{Y} \equiv Y(\mathfrak{gl}(n|m))$ known as the *Yangian*. The $\mathbf{L}$-operator from Eq. (8) is in this context interpreted as a $\mathcal{Y}$-valued matrix on $\mathbb{C}^{n+m}$ which admits the resolution

$$\mathbf{L}(z) = (-1)^{ab+b} E^{ab} \otimes \mathbf{L}^{ab}(z). \qquad (10)$$

The class of solutions to the nonequilibrium protocol considered in this work turn out to be related to certain degenerate representations of $\mathcal{Y}$ which are discussed in Section 4.

The presence of non-trivial grading can be seen as a diagonal 'metric tensor' $\theta$ on the two-particle space $\mathbb{C}^{n+m} \otimes \mathbb{C}^{n+m}$,

$$\theta_{ac,bd} = (-1)^{ab} \delta_{ac} \delta_{bd}, \qquad (11)$$

which allows for an alternative interpretation of Eq. (8) as a braided Yang–Baxter equation on a non-graded vector space $\mathbb{C}^{n+m} \otimes \mathbb{C}^{n+m}$,

$$R^{n|m}(z_1 - z_2)\,\theta\,\big(\mathbf{L}(z_1) \otimes 1\big)\,\theta\,\big(1 \otimes \mathbf{L}(z_2)\big) = \big(1 \otimes \mathbf{L}(z_2)\big)\,\theta\,\big(\mathbf{L}(z_1) \otimes 1\big)\,\theta\,R^{n|m}(z_1 - z_2). \qquad (12)$$

The graded permutation can be expressed in terms of the non-graded permutation $P^{n+m}$ on $\mathbb{C}^{n+m}$ as $P^{n|m} = \theta P^{n+m}$.

The central object of the algebraic Bethe Ansatz solution of integrable quantum models is the fundamental transfer matrix $T_{\square}^{n|m}(z)$ operating on a $N$-particle physical space $(\mathbb{C}^{n|m})^{\otimes N}$ and satisfying the involution property (additional information can be found in appendix B)

$$T_{\square}^{n|m}(z) = \mathrm{Str}_{\mathcal{V}_{\square}^{n|m}}\, \mathbf{L}_{\square}(z) \otimes \cdots \otimes \mathbf{L}_{\square}(z), \qquad \big[T_{\square}^{n|m}(z), T_{\square}^{n|m}(z')\big] = 0. \qquad (13)$$

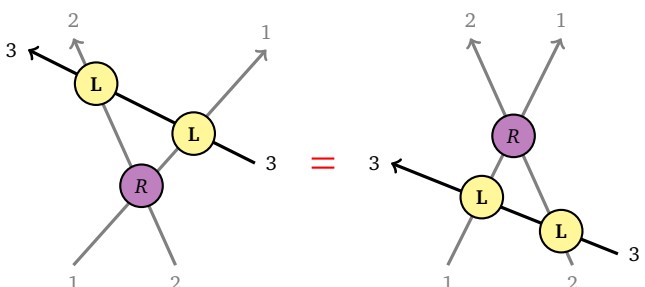

Figure 1: Graphical representation of the Yang–Baxter relation (8). From the particle scattering perspective, the relation imposes the equivalence of two apriori district ways of three consecutive pairwise elastic scatterings. Time direction for physical particles flows vertically and is shown by gray trajectories. The $R$-matrix $R(z_1 - z_2)$ acts proportionally to a graded permutation on two fundamental physical particles in a $\mathfrak{su}(n|m)$ symmetric integrable quantum chain. Lax operators $\mathbf{L}(z_i - z_3)$ on the other hand govern scattering between physical particles with rapidities $z_i$, $i \in \{1, 2\}$, and a fictitious particle carrying rapidity $z_3$ whose time-direction runs horizontally. Graded Yangians $Y(\mathfrak{gl}(n|m))$ are infinite-dimensional associative algebras for the operator-valued coefficients of the $\mathbf{L}$-operator, with the Yang–Baxter equation on $\mathbb{C}^{n|m} \otimes \mathbb{C}^{n|m}$ taking the role of its defining relations.

Commutativity of transfer matrices is ensured by the existence of the $R$-matrix $R^{n|m}_{\square,\square}(z)$ which intertwines two fundamental *auxiliary* representations $\mathcal{V}^{n|m}_{\square}$, i.e. it solves the corresponding (graded) Yang–Baxter relation. We need to emphasize however that Eq. (8) is written with the opposite identification of physical and auxiliary degrees of freedom with respect to the form which is most commonly used in the (algebraic) Bethe ansatz technique. The upshot is that our construction necessitates generic auxiliary modules and not the conventional fundamental auxiliary representations. Indeed, the ordinary set of transfer matrices $T^{n|m}_{\square}(z)$ and their fused counterparts which correspond to finite-dimensional auxiliary irreducible representations of $\mathfrak{gl}(n|m)$ are *unitary* objects and in fact have no natural place in our application.

**Differential Yang–Baxter relation.** Taking the derivative of Eq. (8) with respect to $z = z_1 - z_2$ yields the *differential Yang–Baxter relation* (sometimes also called the Sutherland relation, cf. [74, 81, 94, 95]),

$$\left[ h^{n|m}, \mathbf{L}(z) \circledast \mathbf{L}(z) \right] = \mathbf{L}(z) \circledast \mathbf{L}'(z) - \mathbf{L}'(z) \circledast \mathbf{L}(z), \tag{14}$$

using the short-handed notation $\mathbf{L}'(z) \equiv \partial_z \mathbf{L}(z)$. Equation (14) is simply a consequence of the fact that $\mathfrak{su}(n|m)$-symmetric Hamiltonian densities $h^{n|m}$ coincide with graded permutations $P^{n|m}$ over $\mathbb{C}^{n|m} \otimes \mathbb{C}^{n|m}$, i.e.

$$h^{n|m} = P^{n|m} \partial_z R^{n|m}(z) = P^{n|m}. \tag{15}$$

For the so-called rational spin chains, relation (14) is in fact a simple corollary the zero-curvature property of the Lax connection.[3] What is more important is that the differential Yang–Baxter relation (16) is satisfied on a purely *algebraic* level, i.e. irrespective of representations of the auxiliary components of the $\mathbf{L}$-operator. A general solution to Eq. (15) is given by an operator $\mathbf{L}_{\Lambda_{n+m}}(z)$ acting on a product space of a local physical space and an arbitrary auxiliary representation, that is $\mathcal{V}_{\square} \otimes \mathcal{V}^+_{\Lambda_{n+m}}$. Here $\mathcal{V}^+_{\Lambda_{n+m}}$ denotes a generic *irreducible highest-weight*

---

[3] Relation (14) should not be confused with the lattice version of the Lax representation which takes the local form $\partial_t \mathbf{L}_i(z) = \mathrm{i}[H, \mathbf{L}_i(z)] = \mathbf{A}_{i+1}(z)\mathbf{L}_i(z) - \mathbf{L}_i(z)\mathbf{A}_i(z)$, with matrices $\mathbf{L}_i(z)$ and $\mathbf{A}_i(z)$ corresponding to the spatial and temporal component of the (discrete) connection of the associated auxiliary linear problem.

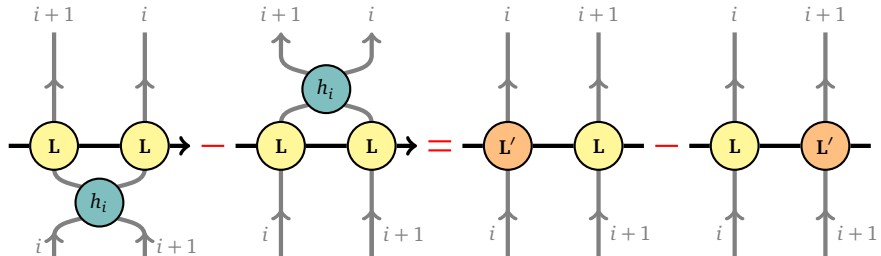

Figure 2: The differential Yang–Baxter equation (see Eq. (14)) takes the form of an operator-valued divergence condition on a one-dimensional lattice. The left-hand side is a schematic representation of the local action of the unitary propagator $\partial_t \Omega_N \simeq [H, \Omega_N]$ which produces a telescopic sum of terms with a single 'defect operator' which coincides with the derivative of the **L**-operator (shown in orange).

representation of $\mathfrak{gl}(n|m)$ Lie superalgebra, characterized by a set of Dynkin labels $\Lambda_{n+m}$ (cf. appendix B).

A key property of algebraic relation (14) is that it remains intact under fusion of auxiliary spaces. This readily makes it possible to extend it to composite (many-particle) auxiliary spaces, namely we may quite generally consider multi-component Lax operators of the following form

$$\mathbb{L}_{\boldsymbol{\Lambda}}(\mathbf{z}) \equiv \mathbf{L}_{\Lambda_{n+m}^1}(z_1) \otimes \mathbf{L}_{\Lambda_{n+m}^2}(z_2) \otimes \cdots \otimes \mathbf{L}_{\Lambda_{n+m}^\ell}(z_\ell), \tag{16}$$

acting on $\mathcal{V}_\square \otimes \mathcal{H}_{\mathrm{aux}}$, with $\mathcal{H}_{\mathrm{aux}}$ representing an arbitrary $\ell$-component auxiliary product space $\mathcal{H}_{\mathrm{aux}} \cong \mathcal{V}_{\Lambda_{n+m}^1} \otimes \cdots \otimes \mathcal{V}_{\Lambda_{n+m}^\ell}$ characterized by a set of weight vectors $\boldsymbol{\Lambda} \equiv \{\Lambda_{n+m}^1, \ldots, \Lambda_{n+m}^\ell\}$ and a vector of complex parameters $\mathbf{z} \equiv \{z_1, \ldots, z_\ell\}$. It is worthwhile emphasizing at this point that the tensor product in Eq. (16) is written with respect to auxiliary spaces $\mathcal{V}_{\Lambda_{n+m}}$, and thus differs from the tensor product of two Lax operator from Eq. (8) which multiplies two copies of local physical (fundamental) spaces $\mathbb{C}^{n|m}$. The multi-component Lax operator $\mathbb{L}_{\boldsymbol{\Lambda}}(\mathbf{z})$ obeys an analogue of Eq. (14), where the $z$-derivative acting on $\mathbf{L}_{\Lambda_{n+m}}(z)$ should be replaced by the chain-rule derivation $\partial_{\mathbf{z}} \equiv \sum_{i=1}^{\ell} \partial_{z_i}$ on $\mathbb{L}_{\boldsymbol{\Lambda}}(\mathbf{z})$, as illustrated in Figure 4.

### 3.2 Amplitude factorization

We consider an $N$-site quantum system with the Hamiltonian $H^{n|m} = \sum_{i=1}^{N-1} h_i^{n|m}$ and impose open boundary conditions. This class of models describes integrable quantum chains symmetric under $\mathfrak{su}(n|m)$ Lie superalgebra [96, 97] whose interactions take a simple form[4]

$$h^{n|m} = (-1)^b E^{ab} \circledast E^{ba}. \tag{17}$$

We adopt the convention for summing over repeated indices throughout the text, unless stated otherwise.

In the boundary-driven setting, the Lindblad dissipator $\mathscr{D}$ gets naturally split into two *independent* incoherent processes assigned to the boundaries of the chain, i.e. $\mathscr{D} = \mathscr{D}_{\mathrm{L}} + \mathscr{D}_{\mathrm{R}}$, where $\mathscr{D}_{\mathrm{L}}$ ($\mathscr{D}_{\mathrm{R}}$) operates only on the first (last) site of the chain. It is perhaps not too surprising that the fixed-point solutions $\rho_\infty$ to Eq. (4) for some bulk Hamiltonian $H^{n|m}$ with *generic* Lindblad boundary dissipators typically yield density matrices lacking any obvious structure. Remarkably however, there exist a class of dissipative boundary conditions for which one may derive an exact algebraic expression for it. Before presenting the precise form of such integrable dissipative boundaries in section 3.3, we first wish to explain why localizing the dissipators to the

---

[4]Here and throughout the text we afforded an unambiguous abuse of notation and replaced all parities $p(a)$ in the superscripts by their argument $a$, i.e. wrote simply $(-1)^{p(a)} \to (-1)^a$.

chain boundaries plays a vital role in our construction and briefly comment on some important consequences. In simple terms, attaching the dissipators only to the boundaries manifestly ensures that the unitary part of the fixed-point condition (4) preserves $\rho_\infty$ up to some residual terms which stick at the boundary sites of the chain. This neat property motivates to use the algebra of (possibly non-local) commuting operators associated to the Hamiltonian $H^{n|m}$ as a trial space of operators for constructing an appropriate $\Omega$-amplitude introduced in Eq. (18). In other words, we shall assume that the steady-state solution of our problem has a well-defined local structure which is related to the symmetry algebra of the Hamiltonian. As explained below, the global symmetry gets broken only due to a mismatch in the boundary conditions, which is essentially the reason why for the steady state, $[H, \rho_\infty] \neq 0$.

We now proceed by employing the following *amplitude factorization* of the density operator $\rho_\infty$,

$$\rho_\infty = \Omega_N \, \Omega_N^\dagger. \tag{18}$$

Let us immediately stress that even though such a decomposition can be applied quite generally, it plays no fundamental role without imposing further restrictions on the amplitude operators $\Omega_N$.[5] Indeed, the factorization property has been originally observed already in the seminal paper [73], where it is referred to as the 'reverse many-body Cholesky factorization'. However, in the class of solutions considered here, $\Omega_N$ need not be a Cholesky factor of a steady state $\rho_\infty$, namely there is no requirement that $\Omega_N$ takes a triangular form when expanded in the standard many-body computational basis of unit matrices spanning $(\mathbb{C}^{m+n})^{\otimes N}$. Nonetheless, since the entire class of solutions which are presented below extends the simplest $\mathfrak{su}(2)$ model to higher dimensional quantum spaces, we shall adopt the factorization property as a starting point of our presentation.[6]

Following the above reasoning, the local symmetry of model is manifestly realized by introducing the following homogeneous *fermionic matrix-product operator*

$$\Omega_N(\mathbf{g}) = \langle \text{vac} | \mathbf{L}(\mathbf{g}) \circledast \mathbf{L}(\mathbf{g}) \circledast \cdots \circledast \mathbf{L}(\mathbf{g}) | \text{vac} \rangle, \tag{19}$$

acting on an $N$-site quantum chain, with symbol $\circledast$ designating the graded tensor product which takes into account the presence of fermionic states. In the pictorial representation, the amplitude represents the lower leg in Figure 3. The key properties of the amplitude operator are:

- Each tensor factor in Eq. (19) is assumed to be a $\mathfrak{gl}(n|m)$-invariant Lax operator parametrized by a continuous real parameter $\mathbf{g}$ (being the coupling strength parameter associated the Lindblad dissipator $\mathscr{D}$). The $\mathbf{L}$-operator acts (by definition) on a local physical space $\mathbb{C}^{n|m}$ and an auxiliary Hilbert space which is at the moment left unspecified and can be thought of as a generic representation of the underlying quantum algebra.

- We have introduced the boundary state $|\text{vac}\rangle$ which will be subsequently referred to as the (auxiliary) *vacuum*. The vacuum state is determined by choice by integrable dissipative boundaries. In all the instances addressed in this work, $|\text{vac}\rangle$ is simply an 'empty state', i.e. a product of highest- or lowest-weight vectors from the irreducible components which form a representation of the auxiliary algebra of the $\mathbf{L}$-operator. This uniquely fixes the vacuum state once the representation labels (i.e. Dynkin labels and additional labels to specify the types of modules involved) associated to the $\mathbf{L}$-operator

---

[5]Notice that, for instance, the factorization property (18) is not gauge-invariant as it notably exhibits a unitary freedom of square roots, i.e. $\Omega \to \Omega U$ for some unitary matrix $U$.

[6]While the factorization property can be sometimes inferred by inspecting the structure of exact solutions found by symbolic algebra routines for small enough instances, its origin and physical significance remains elusive at the moment.

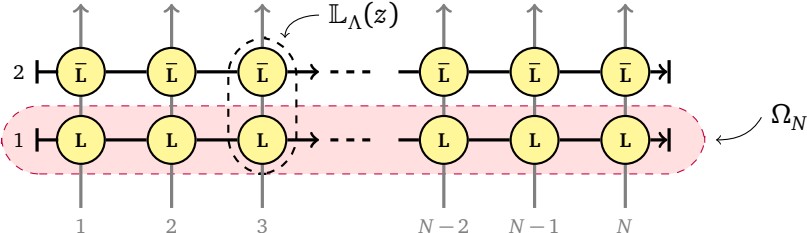

Figure 3: Matrix-product representation of the non-equilibrium steady state $\rho_\infty = \Omega_N \Omega_N^\dagger$: the amplitude operator $\Omega_N$ is represented by the degrees of freedom residing in the bottom row (shown in pink), while its conjugate transpose $\Omega_N^\dagger$ corresponds to the upper row. In terms of an auxiliary scattering process, auxiliary particles are depicted by black lines and propagate in the horizontal direction. They can be viewed as fictitious particles composed of canonical bosons, fermions of complex (super)spins, emanating from the auxiliary vacuum on one end and absorbed by the same vacuum at the other end. Physical degrees of freedom (shown by gray vertical lines) are on the other hand associated to $N$ fundamental particles of $\mathfrak{gl}(n|m)$ Lie superalgebra. The off-shell steady-state density operator admits an interpretation as a vacuum contraction of a homogeneous two-row monodromy operator $\mathbb{M}_\Lambda(z) = \mathbb{L}_\Lambda(z) \circledast \mathbb{L}_\Lambda(z) \circledast \cdots \circledast \mathbb{L}_\Lambda(z)$, where $\mathbb{L}_\Lambda(z) = \mathbf{L}_\Lambda(z) \otimes \bar{\mathbf{L}}_\Lambda(z)$ is a Lax operator which acts on each vertical rung.

from Eq. (19) are being specified. The role of the auxiliary vacua shall be more carefully explained in Section 5 where we treat a few explicit instances.

Let us now return to the differential Yang–Baxter relation. Algebraic property (14) can be readily extended to the entire spin chain Hilbert space $\mathscr{H} \cong (\mathbb{C}^{n+m})^{\otimes N}$ by simply expanding out the commutator $[H^{n|m}, \Omega_N]$ and iteratively applying Eq. (14) at every pair of adjacent lattice sites. This results in a telescoping cancellation mechanism which globally almost annihilates the unitary part of the evolution generated by $\mathscr{L}_0$, leaving behind only residual boundary terms which are an artefact of open boundary conditions. This can be formally expressed in the form [74, 98]

$$\left[H^{n|m}, \Omega_N(\mathbf{g})\right] = \Xi_{\mathrm{L}} \otimes \Omega_{N-1}(\mathbf{g}) - \Omega_{N-1}(\mathbf{g}) \otimes \Xi_{\mathrm{R}}, \tag{20}$$

which can be viewed as the global version of the local condition (14) after contracting with the vacuum $|\mathrm{vac}\rangle$ at the end. We have written $\Xi_{\mathrm{L,R}}$ to denote a pair of 'boundary defect operators' acting only in the boundary particle spaces (their explicit form is not of our interest).

Factorization property (18) indicates that the auxiliary Hilbert space $\mathscr{H}_{\mathrm{aux}}$ associated to the matrix-product representation of the steady-state density matrix $\rho_\infty$ is a two-fold product of auxiliary subspaces which belong to mutually conjugate realizations of the underlying symmetry. Therefore, setting $\ell = 2$ in Eq. (16) and writing shortly $\Lambda_{m+n} \equiv \Lambda$, we arrive at the 'off-shell' representation[7] for $\rho_\Lambda(\mathbf{z})$,

$$\rho_\Lambda(z) = \langle\!\langle \mathrm{vac} | \mathbb{L}_\Lambda(z) \circledast \mathbb{L}_\Lambda(z) \circledast \cdots \mathbb{L}_\Lambda(z) | \mathrm{vac} \rangle\!\rangle, \tag{21}$$

where

$$\mathbb{L}_\Lambda(z) = \mathbf{L}_\Lambda(z) \otimes \bar{\mathbf{L}}_\Lambda(z) \tag{22}$$

is a two-row Lax operator of the form which is represented in Figure 3 by a vertical rung. Similarly, the boundary state $|\mathrm{vac}\rangle\!\rangle$ represents a factorizable state of two auxiliary vacua, $|\mathrm{vac}\rangle\!\rangle = |\mathrm{vac}\rangle \otimes |\mathrm{vac}\rangle$. The internal structure of the vacuum state $|\mathrm{vac}\rangle$, which depends on the rank of symmetry algebra and the choice of integrable boundaries, will be detailed out in Section 5.

---

[7] An 'off-shell' operator is referred to an object of an appropriate algebraic form which is *not required* to be a solution of the fixed-point condition (4).

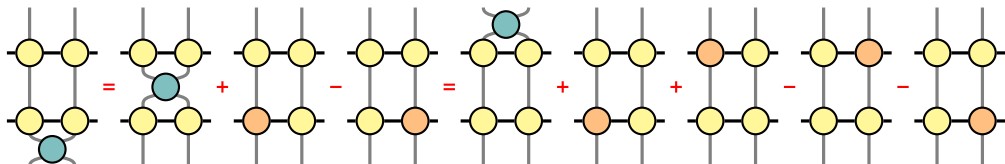

Figure 4: A schematic depiction of $h^{n|m}(\mathbb{L}_\Lambda(z) \circledast \mathbb{L}_\Lambda(z))$, representing the local action of the Hamiltonian $H^{n|m}$ on the density operator $\rho_\infty = \Omega_N \Omega_N^\dagger$ (the coloring adopted from Figure 2, and spectral and representation parameters are suppressed for clarity). The process of brining the interaction $h^{n|m}$ across the horizontal legs generates terms which can be interpreted as a operator divergence condition for two-row Lax operators $\mathbb{L}_\Lambda(z)$.

### 3.3 Boundary compatibility condition

Given the Hamiltonian $H^{n|m}$ and the boundary dissipators $\mathscr{D}$, the fixed-point condition (4) imposes a certain type of bulk-boundary matching condition. It can be inferred from expression (20) that the fixed point condition $\mathscr{L} \rho_\infty = 0$ admits a solution $\rho_\infty$ if and only if there exist an $\Omega$-amplitude (which amounts to find the **L**-operator and the vacuum state $|\text{vac}\rangle$) for which the dissipator $\mathscr{D}$ exactly cancels out the right hand-side of Eq. (20). By plugging in a trial off-shell density operator $\rho_\Lambda(z)$ and demanding the *on-shell* condition one obtains a system of boundary algebraic equations for the undetermined representation parameters which depends also on the physical coupling parameters **g** of the reservoirs. The solution, when it exists, singles out a unique density operator $\rho_\infty(\mathbf{g}(\Lambda, z))$.

Combining Eq. (21) with a general solution of the bulk condition (16) results in two decoupled sets of *boundary compatibility conditions*, which can be cast in the compact form [82]

$$
\begin{aligned}
\langle\!\langle \text{vac}| \left( \mathscr{D}_{\text{L}} + i\partial_z \right) \mathbb{L}_\Lambda(z) &= 0, \\
\left( \mathscr{D}_{\text{R}} - i\partial_z \right) \mathbb{L}_\Lambda(z) |\text{vac}\rangle\!\rangle &= 0.
\end{aligned}
\tag{23}
$$

The boundary conditions of this form generically yield an *overdetermined* system of equations for the free parameters of the two-row Lax operator $\mathbb{L}_\Lambda(z)$. Indeed, it is not difficult to confirm that in spite of integrability of the bulk interactions *generic* boundary dissipators do not lead to any solutions of Eqs. (23). In other words, for some general choice of boundary dissipators there exist no off-shell operator $\rho_\Lambda(z)$ which would satisfy the fixed-point condition of Eq. (4). Of course this should not be surprising at all since typical dissipation processes result in a 'non-integrable' Liouvillian dynamics in which a naïve separation of bulk and boundary parts cannot be justified. Needless to say that in such a case there exists no obvious explicit representation of the steady states either. It is therefore quite remarkable that integrable lattice models with $\mathfrak{su}(n|m)$-symmetric interactions $h^{n|m}$ do allow for certain elementary (so-called integrable) boundary dissipators which lead to non-trivial solutions to boundary equations (23).

### 3.4 Integrable dissipative boundaries

We consider a pair of dissipative boundary processes which involves an arbitrary pair of states from the local Hilbert space $\mathbb{C}^{n|m}$. Denoting them by $|\alpha\rangle$ and $|\beta\rangle$, we posit the jump operators of the form[8]

$$
A_1 = \sqrt{g}\, E_1^{\alpha\beta}, \qquad A_N = \sqrt{g}\, E_N^{\beta\alpha},
\tag{24}
$$

---

[8]In principle the left and right reservoirs can be assigned unequal couplings without spoiling integrability (see e.g. [80]). In this work we prefer for simplicity to concentrate to the situation with equal coupling rates.

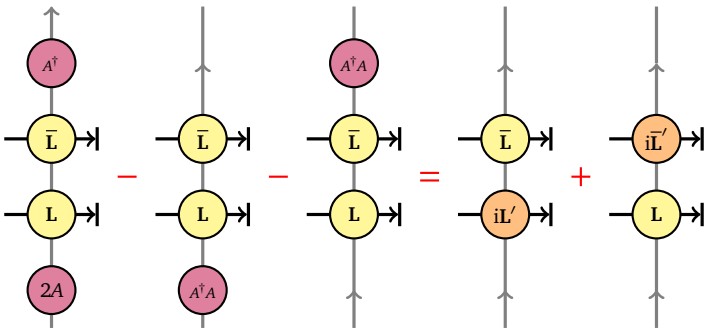

Figure 5: Graphical interpretation of the boundary compatibility condition as given by equation (23) displayed for the right boundary at lattice site $N$. The left-hand side shows schematically the action of the dissipator $\mathscr{D}$ on the $\mathbb{L}$-operator decomposed into three terms which the define the action of the jump operator $A_N$. The termination point of the horizontal arrow signifies the contraction with the right auxiliary vacuum state. Note that the boundary condition has to be satisfied for all values of physical indices.

parametrized by a single reservoir coupling parameter $g$. Since Lindblad dissipators which enter in Eq. (24) operate non-trivially only on the boundary sites of the chain, the jump operator from Eq. (24) can be interpreted as a source and drain associated to $U(1)$ particle currents.

In models with multiple states per site such as $\mathfrak{su}(n|m)$ chains considered here, the diagonal 'density operators' $E_i^{aa}$ obey the following local continuity equations

$$\partial_t \left( E_i^{aa} - E_i^{bb} \right) = \mathrm{i} \left[ H, E_i^{aa} - E_i^{bb} \right] = j_{i-1,i}^{ab} - j_{i,i+1}^{ab}, \tag{25}$$

where $j_i^{ab}$ denote partial currents between two levels $|a\rangle$ and $|b\rangle$ locally at lattice site $i$. Total current densities between two adjacent lattice sites are then obtained by summing over all partial currents, that is $j_{i,i+1}^{a} \equiv \sum_{b=1}^{n} j_{i,i+1}^{ab}$, and fulfil

$$\partial_t E_i^{aa} = \mathrm{i} \left[ H, E_i^{aa} \right] = j_{i-1,i}^{a} - j_{i,i+1}^{a}. \tag{26}$$

Integrable $\mathfrak{su}(n|m)$ symmetric Hamiltonians $H^{n|m}$ conserve the total particle numbers $N^a = \sum_i E_i^{aa}$ independently, i.e. $[H^{n|m}, N^a] = 0$. The addition of dissipation however destroys the conservation of $N^a$ if $a \in \{\alpha, \beta\}$.

To better examine this situation, we notice that dissipative boundaries given by Eq. (24) allow to decompose Liouvillian dynamics into invariant subspaces,

$$\mathscr{H} = \bigoplus_{\underline{v}} \mathscr{H}^{\underline{v}}, \qquad \underline{v} \equiv \{v_1, \ldots, v_{n+m}\} \setminus \{v_\alpha, v_\beta\}, \tag{27}$$

where orthogonal Hilbert subspaces $\mathscr{H}^{\underline{v}}$ are defined via $N^\gamma \mathscr{H}^{\underline{v}} = v_\gamma \mathscr{H}^{\underline{v}}$, with eigenvalues $v_\gamma \in \{0, 1, \ldots, N\}$ for $\gamma \in \{1, \ldots, n + m\} \setminus \{\alpha, \beta\}$. Accordingly, we introduce endomorphisms $\mathcal{O}^{\underline{v}} = \mathrm{End}(\mathscr{H}^{\underline{v}})$, i.e. linear spaces of operators operating on $\mathscr{H}^{\underline{v}}$. This means that states from $\mathscr{H}^{\underline{v}}$ have well-defined values of all particle number operators $N^\gamma$. When $\mathrm{rank}(\mathfrak{g}) > 1$, there exist *at least one* number operator $N^\gamma$ such that

$$\left[ H^{n|m}, N^\gamma \right] = \left[ A_1, N^\gamma \right] = \left[ A_N, N^\gamma \right] = 0. \tag{28}$$

This is an example of the so-called '*strong Liouvillian symmetry*' [99]. In fact, all $N^\gamma$ correspond to strong symmetries, with the exception of the two distinguished indices which belong to a pair of levels affected by the boundary dissipation, that is $\gamma \in \{\alpha, \beta\}$. This immediately implies

degeneracy[9] of the steady states (cf. [82]) and vanishing current expectation values $\langle j^\gamma \rangle_\infty = 0$. Thus only current densities $\langle j^\alpha \rangle_\infty$ and $\langle j^\beta \rangle_\infty$ can take non-vanishing steady-state expectation values.

When dealing with degenerate null spaces of the generator $\mathscr{L}$, the steady state operator $\rho_\infty$ naturally decomposes in terms of independent fixed-point components $\rho_\infty^{\underline{\mu}}$ from individual invariant subspaces $\mathscr{O}^{\underline{\mu}}$, with $\underline{\mu} = \underline{\nu} \setminus \{\nu_\alpha, \nu_\beta\}$. That is, we have

$$\rho_\infty = \sum_{\underline{\mu}} \rho_\infty^{\underline{\mu}}, \qquad \rho_\infty^{\underline{\mu}} = \mathscr{P}^{\underline{\mu}} \rho_\infty, \tag{29}$$

with $\mathscr{P}^{\underline{\mu}}$ denoting orthogonal projectors onto subspaces $\mathscr{O}^{\underline{\mu}}$. Because each invariant component $\rho_\infty^{\underline{\mu}}$ satisfies $\mathscr{L} \rho_\infty^{\underline{\mu}} = 0$ for all values of $\underline{\mu}$, they may be combined in a convex-linear combination

$$\rho_\infty = \sum_{\underline{\mu}} c_{\underline{\mu}} \rho_\infty^{\underline{\mu}}, \tag{30}$$

with $c_{\underline{\mu}}$ representing a $(n + m - 2)$-component vector with non-negative components. The steady-state operator $\rho_\infty$ as defined by Eq. (30) can be thus regarded as a grand canonical nonequilibrium ensemble with coefficients $c_{\underline{\mu}}$, which play the role of particle chemical potentials. Notice that Eq. (30) can be conveniently cast in the form of a matrix-product operator along the lines of ref. [82] for the $\mathfrak{su}(3)$ chain.

Analysing 'integrable boundaries' of Eq. (24) by e.g. computing exact solutions for quantum chains of small length reveals that the fixed point is a non-trivial current-carrying steady state of particularly simple structure. In the remainder of the paper we demonstrate that the steady-state solutions exhibit a particular algebraic representation which directly links to fundamental objects of quantum algebras. It is worthwhile stressing nonetheless that, despite the simplicity of our effective reservoirs, the entire spectrum of $\mathscr{L}$ – typically referred to as the *Liouville decay modes* – remains highly complex and lack any obvious structure. This means that only the fixed-point solutions $\rho_\infty$ of Eq. (4) admit an exact description. It is also instructive to remark here that even the integrable steady state density operators themselves do not enjoy the full quantum group symmetry of the Hamiltonian. Indeed, as a consequence of the foliation (27) of the Lindbladian flow, the global residual symmetry of $\rho_\infty$ is merely $U(1)^{\otimes n+m-2}$. However, as subsequently demonstrated, the *local* symmetry of the $\Omega$-amplitude is much larger. The symmetry content of the steady state solution will be carefully examined in the next sections. Particularly, the local symmetry of the $\Omega$-amplitudes will become apparent on the basis of previously discussed Lax representation (see also Section 6 for additional remarks).

# 4 Graded Yangians

Yangians are certain infinite-dimensional quadratic associative algebras which belong to a class of (quasi-triangular) Hopf algebras, widely referred to as quantum groups. Yangians can be defined in various equivalent ways [101–103]. Here we employ the 'FRT realization' [104] (also known as the 'RTT realization'), in which Yang–Baxter equation (8) takes a role of the defining relation.

We specialize the discussion to Yangians $\mathscr{Y} \equiv Y(\mathfrak{g})$ of Lie superalgebras $\mathfrak{g} = \mathfrak{gl}(n|m)$ [105, 106]. Recall that the signature $n|m$ indicates that the local Hilbert space consists of $n$ bosonic

---

[9]Uniqueness of the steady-state components from individual conserved subspaces follows from the theorem of Evans [100].

and $m$ fermionic states. Generators of $\mathcal{Y}$ are given as the operator-valued coefficients of the Lax operator $\mathbf{L}(z)$ expanded as a formal Laurent series

$$\mathbf{L}^{ab}(z) = \mathbf{L}^{ab}_{(0)} + z^{-1}\,\mathbf{L}^{ab}_{(1)} + z^{-2}\,\mathbf{L}^{ab}_{(2)} + \dots. \tag{31}$$

By imposing Yang–Baxter equation (8) as the defining relation, we obtain an infinite set of quadratic algebraic conditions

$$\left[\mathbf{L}^{ab}_{(r)}, \mathbf{L}^{cd}_{(s)}\right] = (-1)^{ab+ac+bc} \sum_{i=1}^{\min(r,s)} \left(\mathbf{L}^{cb}_{(r+s-i)}\mathbf{L}^{ad}_{(i-1)} - \mathbf{L}^{cb}_{(i-1)}\mathbf{L}^{ad}_{(r+s-i)}\right). \tag{32}$$

The level-0 generators $\mathbf{L}^{ab}_{(0)}$ are scalars belonging to the center of $\mathcal{Y}$. In the scope of our application, we shall only be interested in the class of fundamental *rational* solutions of Eq. (9) which are of degree one in the spectral parameter $z$,[10]

$$\mathbf{L}^{ab}(z) = \mathbf{L}^{ab}_{(0)} + z^{-1}\mathbf{L}^{ab}_{(1)}, \qquad \mathbf{L}^{ab}_{(k)} \equiv 0 \quad \text{for} \quad k \geq 2. \tag{33}$$

This choice represents, in mathematical terms, an evaluation homomorphism from the Yangian to the universal enveloping algebra of $\mathfrak{g}$, $\mathcal{Y}(\mathfrak{g}) \mapsto \mathcal{U}(\mathfrak{g})$. With this restriction, representations of $\mathcal{Y}$ are in one-to-one correspondence with representations of the classical Lie (super)algebra $\mathfrak{g}$.

**Automorphisms.** It is instructive to shorty discuss the gauge freedom due to automorphisms of $\mathcal{Y}$, i.e. transformations which preserve the algebra (32) (cf. [86]). These comprise of (i) rescaling $\mathbf{L}(z)$ with an arbitrary complex-valued scalar function $f(z)$, (ii) shifting the spectral parameter $z \to z + z'$ and (iii) applying a $(n+m)$-dimensional $GL(n|m)$ gauge transformations which acts in $\mathbb{C}^{n|m}$ and is given by two arbitrary invertible matrices $G_L$ and $G_R$, $\mathbf{L}(z) \to G_L\,\mathbf{L}(z)\,G_R$. In addition, there exist anti-automorphisms of $\mathcal{Y}$, i.e. transformations which only preserve the defining relations (8) up to exchanging the order of tensor factors. Examples of these are transposition of the matrix space $\mathbf{L}(z) \to \mathbf{L}^t(z)$, and reflection in the spectral plane $\mathbf{L}(z) \to \mathbf{L}(-z)$. Any composition of two anti-automorphisms is again an automorphism.

**Rank–degenerate realizations.** The list of transformations given above nonetheless does not exhaust all possibilities of realizing $\mathcal{Y}$. As pointed out in [86, 87], equation (8) admits a class of 'degenerate solutions' provided that one relaxes the requirement $\mathbf{L}_{(0)} = 1$. This is a viable choice because the level-0 generators $\mathbf{L}^{ab}_{(0)}$ are central and can therefore take arbitrary (possibly vanishing) values. We may thus quite generally prescribe

$$\mathbf{L}_{(0)} = \mathrm{diag}(\underbrace{1, 1, \dots, 1}_{|I|}, \underbrace{0, \dots, 0}_{|\bar{I}|}), \qquad 1 \leq |I| \leq n + m, \tag{34}$$

modulo equivalent choices which correspond to permutations of 0s and 1s. Such a restriction obviously induces another block structure[11] on $\mathbb{C}^{n|m}$ under which the generators of $\mathcal{Y}$ split as

$$\mathbf{L}_{(1)} = \begin{pmatrix} \mathbf{A}^{ab} & \mathbf{B}^{a\dot{b}} \\ \mathbf{C}^{\dot{a}b} & \mathbf{D}^{\dot{a}\dot{b}} \end{pmatrix}. \tag{35}$$

---

[10]Realizations of $\mathcal{Y}$ which are of higher degree in $z$ have been briefly discussed in [87]. At the moment it remains unclear to us whether these solutions can be important in the studied setup.

[11]We follow the notation of [86, 87] and employ the two-index labelling of the Yangian generators.

The ranges of ordinary (undotted) and dotted indices are

$$a, b \in I = \{1, 2, \ldots, |I| = p + q\}, \qquad \dot{a}, \dot{b} \in \overline{I} = \{1, \ldots, n + m\} \backslash I, \tag{36}$$

where $p$ ($q$) denotes the number of bosonic (fermionic) states in the index set $I$. Similarly, we shall denote by $\dot{p}$ ($\dot{q}$) is the number of bosonic (fermionic) states contained in the complementary set $\overline{I}$. The defining relations of the resulting 'hybrid algebra' $\mathfrak{A}_{n,m}^I$ are readily obtained by plugging Eq. (35) in Eq. (32), and selecting the level-0 generators in accordance with Eq. (34), i.e. $\mathbf{L}_{(0)}^{ab} = \delta_{aI} \delta_{bI}$. Since the generators $\mathbf{D}^{\dot{a}\dot{b}}$ are central, it is convenient to pick a gauge by setting $\mathbf{D}^{\dot{a}\dot{b}} = \delta_{\dot{a}\dot{b}}$. The remaining non-trivial commutation relations read

$$\begin{aligned}
\left[\mathbf{A}^{ab}, \mathbf{A}^{cd}\right] &= (-1)^{ab+ac+bc}(\delta_{ad}\,\mathbf{A}^{cb} - \delta_{cb}\,\mathbf{A}^{ad}), & \left[\mathbf{A}^{ab}, \mathbf{B}^{c\dot{d}}\right] &= -(-1)^{ab+ac+bc}\delta_{cb}\,\mathbf{B}^{a\dot{d}}, \\
\left[\mathbf{A}^{ab}, \mathbf{C}^{\dot{c}d}\right] &= (-1)^{ab+a\dot{c}+b\dot{c}}\delta_{ad}\,\mathbf{C}^{\dot{c}b}, & \left[\mathbf{B}^{a\dot{b}}, \mathbf{C}^{\dot{a}b}\right] &= (-1)^{\dot{a}}\delta_{ab}\,\delta_{\dot{a}\dot{b}}, \tag{37} \\
\left[\mathbf{B}^{a\dot{b}}, \mathbf{B}^{c\dot{s}}\right] &= 0, & \left[\mathbf{C}^{\dot{a}b}, \mathbf{C}^{\dot{c}s}\right] &= 0.
\end{aligned}$$

## 4.1 Oscillator realizations

Commutation relations (37) have been derived in [86, 87], where the authors provide a realization in terms of $\mathfrak{gl}(p|q)$ 'super spin' generators $\mathbf{J}^{ab}$ (for $a, b \in I$, and with $p + q = |I|$),

$$\left[\mathbf{J}^{ab}, \mathbf{J}^{cd}\right] = \delta_{cb}\,\mathbf{J}^{ad} - (-1)^{(a+b)(c+d)}\delta_{ad}\,\mathbf{J}^{cb}, \tag{38}$$

and additional $|I| \cdot |\overline{I}|$ canonical bosonic or fermionic oscillators which obey graded canonical commutation relations

$$\left[\xi^{\dot{a}b}, \overline{\xi}^{c\dot{d}}\right] = \delta_{cb}\,\delta_{\dot{a}\dot{d}}, \tag{39}$$

where a generator $\overline{\xi}^{a\dot{b}}$ should be understood as a creation operator of a bosonic (fermionic) oscillator if $p(\dot{b}) = 0$ ($p(\dot{b}) = 1$), for $a \in I$ and $\dot{b} \in \overline{I}$. The oscillator part of the algebra $\mathfrak{A}_{n,m}^I$, denoted by $\mathfrak{osc}(p+\dot{p}|q+\dot{q})$, is associated with a multi-component Fock space $\mathscr{B}^{\otimes(p+\dot{p})} \otimes \mathscr{F}^{\otimes(q+\dot{q})}$, where each factor $\mathscr{B}$ ($\mathscr{F}$) belongs to an irreducible bosonic (fermionic) Fock space. In terms of these 'super spins' and 'super oscillators', the level-1 generators $\mathbf{L}_{(1)}^{ab}$ take the *canonical* form

$$\begin{aligned}
\mathbf{A}^{ab} &= -(-1)^b\left(\mathbf{J}^{ab} + \mathbf{N}^{ab}\right), \\
\mathbf{B}^{a\dot{b}} &= \overline{\xi}^{a\dot{b}}, \\
\mathbf{C}^{\dot{a}b} &= -(-1)^b \xi^{\dot{a}b}, \tag{40} \\
\mathbf{D}^{\dot{a}\dot{b}} &= \delta_{\dot{a}\dot{b}},
\end{aligned}$$

where

$$\mathbf{N}^{ab} = \sum_{\dot{d} \in \overline{I}} \overline{\xi}^{a\dot{d}} \xi^{\dot{d}b} + \tfrac{1}{2}(-1)^{a+\dot{d}}\delta_{ab}. \tag{41}$$

## 4.2 Partonic Lax operators

For a Lie superalgebra $\mathfrak{g}$ of $\mathrm{rank}(\mathfrak{g}) = n+m$, there are in total $2^{n+m}$ distinct types of hybrid-type subalgebras $\mathfrak{A}_{n,m}^I$. The latter are in a bijective correspondence with all possible choices of set $I$, where the counting excludes the various possibilities of choosing the grading. By additionally excluding permutation equivalent choices, we eventually deal with finite-dimensional Lie superalgebras of the type $\mathfrak{gl}(p|q) \otimes \mathfrak{osc}(p + \dot{p}|q + \dot{q})$.

The simplest rank-degenerate solutions of the graded Yang–Baxter algebra (8) belong to the single-indexed sets $|I| = 1$ and were dubbed in [85] as the *partonic* solutions. These consist solely from $n + m - 1$ oscillators arranged in a distinctive cross-shaped form,

$$
\mathbf{L}_{\{a\}}(z) = \begin{pmatrix}
1 & & & -(-1)^a \xi^{1,a} & & & \\
& \ddots & & \vdots & & & \\
& & 1 & -(-1)^a \xi^{a-1,a} & & & \\
\overline{\xi}^{a,1} & \dots & \overline{\xi}^{a,a-1} & z - \mathbf{N}_{\overline{I}}^a & \overline{\xi}^{a,a+1} & \dots & \overline{\xi}^{a,n+m} \\
& & & -(-1)^a \xi^{a+1,a} & 1 & & \\
& & & \vdots & & \ddots & \\
& & & -(-1)^a \xi^{n+m,a} & & & 1
\end{pmatrix}, \tag{42}
$$

with

$$
\mathbf{N}_{\overline{I}}^a = \sum_{b \in \overline{I}} (-1)^b \left( \overline{\xi}^{ab} \xi^{ba} + \tfrac{1}{2}(-1)^{a+b} \right). \tag{43}
$$

Here the integers $a \in \{1, \dots, n + m\}$ in the subscript of $\mathbf{L}_{\{a\}}(z)$ are being used to indicate the location of the single non-vanishing level-0 generator, cf. Eq. (34). As shown in appendix B, all Lax operators associated to $\mathfrak{A}_{m,n}^{|I| \geq 2}$ can be systematically generated from the partonic solutions which carry $\mathfrak{A}_{n,m}^{|I| = 1}$ with aid of the universal fusion formula, yielding 'multi-partonic' Lax operators which are equivalent to canonical Lax operators given by Eq. (40).

## 5 Exact steady states for integrable quantum spin chains

In this section we finally present a few explicit examples for the steady-state solutions of the boundary-driven $\mathfrak{su}(n|m)$ quantum chains, subjected to integrable dissipative boundaries given by Eq. (24). As the first step, we account for the kinematic constraints and construct off-shell density operators which take the universal form of equation (21). Subsequently, the goal is to find an appropriate internal structure of the Lax operator $\mathbb{L}_\Lambda$ and the auxiliary vacuum state $|\text{vac}\rangle$ which fulfil the requirements of the boundary equations (23).

Notice first that there exist in total $2 \times \binom{n+m}{2}$ ways of assigning the dissipators of Eq. (24), representing all the possibilities of selecting a pair of target levels $|\alpha\rangle$ and $|\beta\rangle$. The extra factor of 2 comes from exchanging $|\alpha\rangle$ with $|\beta\rangle$ which results in a state of the opposite chirality, i.e. the particle currents get reversed. It turns out that every choice of $|\alpha\rangle$ and $|\beta\rangle$ leads to a solution to Eq. (4) which is uniquely characterized by specifying a representation of the auxiliary algebra $\mathfrak{A}_{n,m}^I$ of the Lax operator $\mathbb{L}_\Lambda(z)$.

### 5.1 Fundamental integrable spin models

The significance of partonic Lax operators and the structure of the steady-state solutions is perhaps best illustrated explicitly by working out a few simplest examples. To this end, we first consider the non-graded interactions, and initially examine the most studied case of the $\mathfrak{su}(2)$ spin chain (the isotropic Heisenberg spin-1/2 model), with interaction density $h^{2|0} = \sum_{a,b=1}^{2} E^{ab} \otimes E^{ba}$. Let us remark that this particular instance has been considered initially in the seminal paper [73] where the solution was found with a somewhat different approach, and afterwards re-obtained in a more compact and symmetric form in [74]. The derivation from Yang–Baxter algebra has been presented in [81]. Nevertheless, to uncover the connection with partonic Lax operators and embed this solution in a unified theoretic framework, we shall reproduce it below once again.

In the $\mathfrak{su}(2)$ spin chain, the local building block of the $\Omega$-amplitude is given by a two-parametric Lax operator $\mathbf{L}_j^-(z)$ acting on $\mathscr{V}_\square \otimes \mathscr{V}_j^-$, whose auxiliary space $\mathscr{V}_j^-$ represents a *lowest-weight* $\mathfrak{sl}(2)$ module spanned by an infinite tower of states $\{|k\rangle\}_{k=0}^\infty$. We adopt the $\mathfrak{sl}(2)$ spin generators obeying algebraic relations

$$\left[\mathbf{J}^3, \mathbf{J}^\pm\right] = \pm \mathbf{J}^\pm, \qquad \left[\mathbf{J}^+, \mathbf{J}^-\right] = 2\mathbf{J}^3, \tag{44}$$

whose action on $\mathscr{V}_j^-$ is prescribed by

$$\mathbf{J}^3|k\rangle = (k-j)|k\rangle, \quad \mathbf{J}^+|k\rangle = (2j-k)|k+1\rangle, \quad \mathbf{J}^-|k\rangle = k|k-1\rangle. \tag{45}$$

State $|0\rangle$ has the lowest weight, $\mathbf{J}^-|0\rangle = 0$, and will be referred to as the vacuum.

By recalling that all the solutions factorize in accordance with Eq. (18), the off-shell Lax operator $\mathbb{L}_\Lambda(z)$ which defines the steady-state solution $\rho_\infty$ is a product of two copies of auxiliary representations, cf. Eq. (22). This factorization makes it possible to express the final solution by only specifying a pair of complex parameters: the $\mathfrak{sl}(2)$ weight which is interpreted as a complex spin $j$, and a complex-valued spectral parameter $z$. Specifically, the two-parameteric off-shell Lax operator which we denote by $\mathbb{L}_j(z)$ is represented in the following compact form

$$\mathbb{L}_j(z) = \mathbf{L}_j^{[1]-}(z)\mathbf{L}_{-j}^{[2]+}(z). \tag{46}$$

The notation used is as follows. Integer indices in superscript brackets are used to assign an operator $\mathbf{L}_j(z)$ to a tensor factor in a multi-component auxiliary space. In addition, in the superscripts we also employed extra parity signatures which are required to correctly assign the type of the $\mathfrak{sl}(2)$ module. Namely, while the first auxiliary copy is realized in the lowest-weight module as prescribed by Eq. (45), the second factor in Eq. (46) must be associated with the highest-weight realization of $\mathfrak{sl}(2)$ algebra $\mathscr{V}_j^+$. The highest-weight Lax operator $\mathbf{L}_j^+(z)$ can be readily obtained from $\mathbf{L}_j^-(z)$ by applying the spin-reversal transformation

$$\mathscr{V}_j^- \to \mathscr{V}_j^+: \qquad \mathbf{J}^\pm \to \mathbf{J}^\mp, \qquad \mathbf{J}^3 \to -\mathbf{J}^3. \tag{47}$$

The highest weight state $|0\rangle$ from $\mathscr{V}_j^+$ is distinguished by $\mathbf{J}^+|0\rangle = 0$.

Plugging the off-shell form of Eq. (46) into the boundary conditions (23) yields a system of polynomial equations with the unique solution

$$z = 0, \qquad j = \frac{\mathrm{i}}{g}. \tag{48}$$

Notice that the auxiliary vacuum state takes the product form, $|\text{vac}\rangle\!\rangle = |0\rangle \otimes |0\rangle$, and is determined by the internal structure of $\mathbb{L}_j(z)$. The driving is reversed by exchanging the target states $|\alpha\rangle \leftrightarrow |\beta\rangle$. This amounts to interchange the factors in Eq. (46).

Before proceeding with other examples, let us stress again that a proper identification of the internal structure of the two-leg Lax operator $\mathbb{L}_j(z)$ is crucial. Once the convention for labelling the irreducible $\mathfrak{sl}(2)$ Verma modules is being fixed, there exist only one correct assignment of irreducible spaces (incorrect assignments produce a system of boundary equations which admits no solution).

**Asymmetric driving.** Let us mention a simple trick which enables to generalize the solutions to the case of *unequal* reservoir coupling constants. Considering as an example the $\mathfrak{su}(2)$ spin chain, we may impose an asymmetric pair of Lindblad jump operators of the form

$$A_1 = \sqrt{g/\zeta}\, E_1^{12}, \qquad A_N = \sqrt{g\zeta}\, E_N^{21}. \tag{49}$$

This choice yields an extended class of solutions which is connected to the special case of equal couplings by a diagonal tilting transformation – a one-parameter automorphism of $\mathcal{Y}$ – by applying

$$\mathbf{L}_j^-(z) \to \mathbf{L}_j^-(z) \begin{pmatrix} \zeta & 0 \\ 0 & 1 \end{pmatrix}, \tag{50}$$

on every local spin space $\mathbb{C}^2$. The solution to the boundary compatibility conditions is then given by

$$z = \frac{i}{2}\left(\frac{1}{g\,\zeta} - \frac{\zeta}{g}\right), \qquad j = \frac{i}{2}\left(\frac{1}{g\,\zeta} + \frac{\zeta}{g}\right). \tag{51}$$

**Models of higher-rank symmetry.** The simplest higher-rank model is the $\mathfrak{su}(3)$ spin chain (with interaction $h^3$), often called in the literature as the Lai–Sutherland model [107, 108]. Solutions to the fixed-point condition (4) have been originally identified and parametrized in [82] and now represent a *degenerate* manifold of steady states. As discussed earlier in Section 3.4, degeneracy of the null space of $\mathscr{L}$ is a consequence of the conservation of the number operator $N^\gamma$ associated to a distinguished noise-protected state $|\gamma\rangle$ (which is arbitrary, depending on $\bar{I}$).

The Lax operator for the $\Omega$-amplitude now operates on a space of three auxiliary particles, a *non-compact* $\mathfrak{sl}(2)$ spin and two species of canonical bosons. Bosonic particles obey canonical oscillator algebra,

$$\left[\mathbf{b}, \mathbf{b}^\dagger\right] = 1, \quad \left[\mathbf{h}, \mathbf{b}^\dagger\right] = \mathbf{b}^\dagger, \quad \left[\mathbf{h}, \mathbf{b}\right] = -\mathbf{b}, \quad \mathbf{h} \equiv \mathbf{b}^\dagger \mathbf{b} + \tfrac{1}{2}, \tag{52}$$

and live in the canonical Fock space spanned by a tower of states $\{|k\rangle\}_{k=0}^\infty$. Similarly as in the case of $\mathfrak{sl}(2)$ spins, one has to distinguish two distinct realizations of bosonic Fock spaces $\mathscr{B}^\pm$,

$$\mathscr{B}^+: \qquad \mathbf{b}\,|0\rangle = 0, \quad \mathbf{b}^\dagger\,|k\rangle = |k+1\rangle, \tag{53}$$

$$\mathscr{B}^-: \qquad \mathbf{b}^\dagger\,|0\rangle = 0, \quad \mathbf{b}\,|k\rangle = |k+1\rangle, \tag{54}$$

related to each other by an algebra automorphism

$$\mathscr{B}^+ \to \mathscr{B}^-: \qquad \mathbf{b}^\dagger \to \mathbf{b}, \quad \mathbf{b} \to -\mathbf{b}^\dagger, \quad \mathbf{h} \to -\mathbf{h}, \tag{55}$$

which is interpreted as the particle-hole conjugation.

By assigning the dissipation to states $I = \{1, 2\}$, the off-shell $\Omega$-amplitude is constructed from the Lax operator $\mathbf{L}_{\{1,2\}}(z)$ which carries a representation of algebra $\mathfrak{A}_{3,0}^I \cong \mathfrak{sl}(2) \otimes \mathscr{B} \otimes \mathscr{B}$ and takes the canonical form of Eq. (40),

$$\mathbf{L}_{\{1,2\}}(z) = \begin{pmatrix} z + \mathbf{J}^3 - \mathbf{h}_1 & \mathbf{J}^- - \mathbf{b}_1^\dagger \mathbf{b}_2 & \mathbf{b}_1^\dagger \\ \mathbf{J}^+ - \mathbf{b}_2^\dagger \mathbf{b}_1 & z - \mathbf{J}^3 - \mathbf{h}_2 & \mathbf{b}_2^\dagger \\ -\mathbf{b}_1 & -\mathbf{b}_2 & 1 \end{pmatrix}. \tag{56}$$

Now it suffices to repeat the logic used before on the $\mathfrak{su}(2)$ case, and define a factorized off-shell Lax operator $\mathbb{L}_j^\omega(z)$ for the steady-state solution $\rho_\infty$ in the form

$$\mathbb{L}_j^\omega(z) = \mathbf{L}_j^{[1]\,\omega}(z) \mathbf{L}_{-j}^{[2]\,\overline{\omega}}(-z). \tag{57}$$

This time we equipped each tensor copy with an additional label $\omega$ which, as argued earlier, is needed to supply the information about the types of $\mathfrak{sl}(2)$ and Fock modules. After determining the right $\omega$, the coupling constant $g$ is linked to the free representation parameters $z, j$ through the boundary equations (23) with the solution

$$z = \frac{1}{2}, \qquad j = -\frac{i}{g}, \qquad \omega = (-|-, +). \tag{58}$$

The delimiter in $\omega$ was used to explicitly distinguish the $\mathfrak{sl}(2)$ module $\mathscr{V}_j^{\pm}$ (on the left) from the signatures belonging to the product of Fock spaces $\mathscr{B}^{\pm}$ (on the right, in the ascending order). Specifically, the above instance requires a lowest-weight type $\mathfrak{sl}(2)$ representation and to assign $\mathscr{B}^-$ ($\mathscr{B}^+$) to the first (second) bosonic oscillator.

Before heading on to the more involved examples of fermionic models, let us spent a few more words on the non-trivial structure of the vacuum $|\text{vac}\rangle$ and, in particular, to *inequivalent* roles of the highest and lowest type of (auxiliary) representations. As said earlier, in order to construct an off-shell $\Omega$-amplitude it is first required to infer an appropriate 'internal structure' for the auxiliary space of $\mathbf{L}_{\Lambda_{n+m}}(z)$. Only then it is possible to proceed by solving the corresponding finite system of polynomial equations (23). The upshot here is that the module type labels $\omega$ are essential to assign $\rho_{\infty}$ an appropriate chiral structure. For instance, an incorrect assignment of the auxiliary bosons in expression (56), e.g. by imposing two identical representations $\mathscr{B}^+$, would violate the boundary compatibility conditions. Finally, one can easily verify that the Lax operator $\mathbf{L}_{\{1,2\}}(z)$ is in fact equivalent to the Lax operator found previously in [82].[12]

## 5.2 Fermionic models

In this section we generalize the above construction to the steady state solutions which pertain to graded $\mathfrak{su}(n|m)$ chains, representing the simplest class of interacting integrable models with fermionic degrees of freedom. We retain the dissipative boundaries given by Eq. (24).[13] Besides bosonic oscillators, the auxiliary particle spaces now also involve canonical fermions from two-dimensional spaces $\mathscr{F}$.

The defining $\mathfrak{gl}(1|1)$ representation is spanned by two basis states $|0\rangle$ and $|1\rangle$. The 'highest weight' type representation, denoted by $\mathscr{F}^+$, is prescribed by

$$\mathscr{F}^+: \qquad \mathbf{c}^{\dagger}|1\rangle = 0, \quad \mathbf{c}^{\dagger}|0\rangle = |1\rangle, \quad \mathbf{c}|0\rangle = 0, \quad \mathbf{c}|1\rangle = |0\rangle, \tag{59}$$

where the generators obey canonical anticommutation relations

$$\left[\mathbf{n}, \mathbf{c}^{\dagger}\right] = \mathbf{c}^{\dagger}, \quad \left[\mathbf{n}, \mathbf{c}\right] = -\mathbf{c}, \quad \left[\mathbf{c}, \mathbf{c}^{\dagger}\right] = 1. \tag{60}$$

Similarly, the 'lowest weight' representation $\mathscr{F}^-$ is obtained from $\mathscr{F}^+$ by virtue of the particle-hole mapping

$$\mathscr{F}^+ \to \mathscr{F}^-: \qquad \mathbf{c} \to \mathbf{c}^{\dagger}, \quad \mathbf{c}^{\dagger} \to \mathbf{c}, \quad \mathbf{n} \to 1 - \mathbf{n}. \tag{61}$$

**Free fermions.** Arguably the simplest fermionic integrable system is a tight-binding model of *non-interacting* spinless fermions (with a homogeneous chemical potential) whose interaction is invariant under $\mathfrak{gl}(1|1)$ Lie superalgebra and in terms of canonical fermions reads[14]

$$h_i^{1|1} = c_i^{\dagger} c_{i+1} + c_{i+1}^{\dagger} c_i - n_i - n_{i+1} + 1. \tag{62}$$

---

[12]In order to exactly match the Lax operator from ref. [82] and recover the canonical representation of the Lax operator $\mathbf{L}_{\{1,3\}}$, one should first redefine the algebra generators to eliminate the redundant parameter $\eta$, apply a diagonal gauge transformation with $G_L = 1$, $G_R = \text{diag}(1, 1, -1)$, and ultimately the particle-hole transformations on the auxiliary spin and oscillator species.

[13]A comment on the Jordan–Wigner transformation: When expressed in terms of the fermionic generators, the dissipator attached to the first lattice site differs from its non-graded counterpart by a (non-local) Jordan–Wigner 'string operator' $W$, indicating that fermionization of the boundary-driven spin chain maps to a model of non-local dissipation. This discrepancy between the two formulations which is due to the presence of $W$ is however immaterial as far as only the steady states are of our interest, the reason being that $W$ commutes with both the steady state $\rho_{\infty}$ and the total Hamiltonian $H^{n|m}$.

[14]The interaction $h^{1|1}$ can also be expressed in terms of the fundamental $\mathfrak{su}(2)$ generators. Up to boundary terms this yields the XX model in a homogeneous external field, that is $h^{1|1} = \sigma^+ \otimes \sigma^- + \sigma^- \otimes \sigma^+ + \frac{1}{2}(\sigma^z \otimes 1 + 1 \otimes \sigma^z)$.

In spite of its simplicity, it is remarkable that the corresponding integrable steady states involve auxiliary spaces which belong to *non-canonical* $\mathfrak{gl}(1|1)$ representations.

The fermionized integrable reservoirs provided by Eq. (24) are interpreted as an inflow (outflow) of spinless fermions at the left (right) boundary with rate $g$,

$$A_1 = \sqrt{g}\, E_1^{21} \equiv \sqrt{g}\, c_1^\dagger, \qquad A_N = \sqrt{g}\, E_N^{12} \equiv \sqrt{g}\, c_N. \tag{63}$$

To find the unique solution to the fixed-point condition (4), we follow the procedure from the previous section and first consider the following two partonic Lax elements,

$$\mathbf{L}_{\{1\}}(z) = \begin{pmatrix} z - (\mathbf{n} - \frac{1}{2}) & \mathbf{c}^\dagger \\ -\mathbf{c} & 1 \end{pmatrix}, \qquad \mathbf{L}_{\{2\}}(z) = \begin{pmatrix} 1 & \mathbf{c} \\ \mathbf{c}^\dagger & z + (\mathbf{n} - \frac{1}{2}) \end{pmatrix}. \tag{64}$$

By merging them together using the fusion rule (see appendix B for details) we have

$$\mathbf{L}_\lambda(z) \simeq \mathbf{L}_{\{1\}}^{[1]}(z_+)\,\mathbf{L}_{\{2\}}^{[2]}(z_-), \qquad z_+ = z + \lambda + \tfrac{1}{2}, \quad z_- = z - \lambda + \tfrac{1}{2}. \tag{65}$$

The outcome is a two-parameteric Lax operator $\mathbf{L}_\lambda(z)$ whose auxiliary space is identified with an indecomposable *non-unitary* representation denoted here by $\mathcal{V}_\lambda$. The latter can be realized in terms of canonical generators (60) as

$$\mathbf{L}_\lambda^\pm(z) = \begin{pmatrix} z_+ - \mathbf{n} & -2\lambda\, \mathbf{c}^\dagger \\ -\mathbf{c} & z_- + \mathbf{n} \end{pmatrix}, \qquad \lambda = \tfrac{1}{2}(z_+ - z_-), \tag{66}$$

where the complex representation parameter $\lambda$ is the *central charge*.[15] To distinguish between the highest- and lowest-weight types of representations we shall use an extra superscript label, using the convention that $\mathcal{V}_\lambda^\pm$ is associated with the Fock space $\mathscr{F}^\pm$, i.e. $(\mathbf{L}_\lambda^\pm)^{12}\,|0\rangle = 0$.

In close analogy to the $\mathfrak{su}(2)$ case, the local unit of the amplitude operator $\Omega_N$ is now built from $\mathbf{L}$-operator $\mathbf{L}_\lambda^-(z)$. The undetermined representation parameters are finally obtained from the boundary conditions (24), yielding the unique solution

$$z = \frac{1}{2}, \qquad \lambda = \frac{i}{g}. \tag{67}$$

The factorization property (18) implies that $\rho_\infty(g)$ is constructed from a two-component Lax operator $\mathbb{L}_\lambda(z)$ which explicitly reads

$$\mathbb{L}_\lambda(z) = \mathbf{L}_\lambda^{[1]-}(z)\,\mathbf{L}_\lambda^{[2]+}(-z), \qquad \text{with} \qquad z = \frac{1}{2}, \quad \lambda = \frac{i}{g}. \tag{68}$$

The driving may be reversed by first applying the particle-hole transformation on the physical fermions (see Eq. (63)), exchanging the order of factors in Eq. (68), and ultimately setting

$$\mathbb{L}_\lambda(z) = \mathbf{L}_\lambda^{[1]+}(z)\,\mathbf{L}_\lambda^{[2]-}(-z), \qquad \text{with} \qquad z = \frac{1}{2}, \quad \lambda = -\frac{i}{g}. \tag{69}$$

The auxiliary vacuum state $|\mathrm{vac}\rangle\!\rangle$ remains the product of the lowest- and highest-weight states $|0\rangle$ from the fermionic Fock modules $\mathscr{F}^\pm$.

We find it instructive to remark that the model of free fermions takes a special place among the $\mathfrak{gl}(n|m)$ quantum chains which plays nicely with the fact that the model is compatible with a larger set of integrable boundary dissipators. It is rather remarkable however that such an extended set of solutions still admits the Lax representation, albeit the latter does no longer exhibit the usual additive property. We are not sure whether this enlarged set of steady-state solutions is still related to representation theory of Yangians. Further details and explicit results are presented in appendix C.

---

[15] Verma module $\mathcal{V}_\lambda^\pm$ is a 2-dimensional indecomposable representation of $\mathfrak{gl}(1|1)$ which is unitary only at $\lambda = \frac{1}{2}$ (where it coincides with the Fock space representation of canonical fermions). At $\lambda = 0$ it becomes atypical and reducible.

**Example: SUSY t-J model.** Integrable spin chains whose interactions coincide with graded permutations have been initially considered in [96, 97]. A prominent (and generic) example is the $\mathfrak{su}(1|2)$-symmetric integrable spin chain which is mappable to the t-J model at the 'supersymmetric point' [109] ($2t = J$). The spectral problem of the model has been solved with Bethe Ansatz techniques in [110–114].

The local Hilbert space is now isomorphic to $\mathbb{C}^{1|2}$, and is spanned by a (bosonic) empty state $|0\rangle$ and a pair of spin-carrying electrons, $|\uparrow\rangle \equiv c_\uparrow^\dagger |0\rangle$ and $|\downarrow\rangle \equiv c_\downarrow^\dagger |0\rangle$, representing fermionic states. The interaction density can be expanded in terms of canonical fermions and reads ($\sigma = \uparrow, \downarrow$)

$$h_{i,i+1}^{1|2} = -\mathscr{P}\left(c_{i,\sigma}^\dagger c_{i+1,\sigma} + c_{i+1,\sigma}^\dagger c_{i,\sigma}\right)\mathscr{P} + 2\left(\vec{S}_i \cdot \vec{S}_{i+1} - \tfrac{1}{4}n_j n_{i+1}\right) + n_i + n_{i+1}, \tag{70}$$

where $\vec{S}_i^\alpha = \tfrac{1}{2}c_{i,\sigma}^\dagger \vec{\sigma}_{\sigma,\sigma'} c_{i,\sigma'}$ and the projector $\mathscr{P} = \prod_{i=1}^N (1 - n_{i,\uparrow}n_{i,\downarrow})$ was used to eliminate the forbidden doubly-occupied state $|\uparrow\downarrow\rangle \equiv c_\uparrow^\dagger c_\downarrow^\dagger |0\rangle$.

The grading can be distributed in various ways.[16] We shall adopt $|0| = 0$, and regard the empty state $|0\rangle$ as the highest-weight state (vacuum) in the physical Hilbert space. Then, we may consider one of the following three options,

$$(1) \quad \otimes\!-\!\odot \qquad (2) \quad \odot\!-\!\otimes \qquad (3) \quad \otimes\!-\!\otimes, \tag{71}$$

represented by Kac–Dynkin diagrams.[17] It is important to remark here that the choice of grading is entirely *independent* from the set $I = \{\alpha, \beta\}$ which specifies a pair of states subjected to the dissipators.

Let us first set the grading to $|1| = 0$ and $|2| = |3| = 1$, which corresponds to diagram (1). Incoherent conversion processes induced by the dissipators can be described by any of the following three possibilities:

$$(a) \quad |0\rangle \longleftrightarrow |\uparrow\rangle \qquad (b) \quad |0\rangle \longleftrightarrow |\downarrow\rangle \qquad (c) \quad |\uparrow\rangle \longleftrightarrow |\downarrow\rangle. \tag{72}$$

Options (a) and (b) represent *fermionic* driving and physically correspond to Markovian transitions between two states of opposite parities, namely a spin-carrying electron and the unoccupied state $|0\rangle$. Option (c) is different, and affects the *bosonic* sector (i.e. the $\mathfrak{su}(2)$ doublet) by triggering incoherent spin flips.

Let us first address option (a), corresponding to assigning the following pair of Lindblad jump operators

$$A_1 = \sqrt{g}\left(1 - n_{1,\uparrow}\right)c_{1,\downarrow}, \qquad A_N = \sqrt{g}\left(1 - n_{N,\uparrow}\right)c_{N,\downarrow}^\dagger. \tag{73}$$

This instance pertains to $I = \{1,2\}$, $\bar{I} = \{3\}$, with $p = q = 1$ and $\dot{q} = 1$, which defines the auxiliary algebra $\mathfrak{A}_{1,2}^{\{1,2\}}$ of the product structure $\mathfrak{gl}(1|1) \otimes \mathscr{F} \otimes \mathscr{B}$. The corresponding canonical Lax operator is of the form

$$\mathbf{L}_{\{1,2\}}(z) = \begin{pmatrix} z - \mathbf{J}^{11} - \left(\bar{\xi}^{13}\xi^{31} - \tfrac{1}{2}\right) & \mathbf{J}^{12} + \bar{\xi}^{13}\xi^{32} & \bar{\xi}^{13} \\ -\mathbf{J}^{21} - \bar{\xi}^{23}\xi^{31} & z + \mathbf{J}^{22} + \left(\bar{\xi}^{23}\xi^{23} + \tfrac{1}{2}\right) & \bar{\xi}^{23} \\ -\xi^{31} & \xi^{32} & 1 \end{pmatrix}, \tag{74}$$

---

[16]From the algebraic point of view, distinct inequivalent gradings indicate that Lie superalgebras do not admit unique simple roots. All distinct possibilities are however related under certain boson-fermion duality transformations. In the context of Bethe Ansatz these correspond to inequivalent Bethe vacua and different ways of proceeding to higher levels in the nesting scheme (see e.g. [114–116]).

[17]By convention we draw an open circle if two adjacent states are of the same parity and a crossed circle when their parities differ (assuming $|0| = 0$), while moving from left to right.

where the generators $\mathbf{J}^{ab}$ are associated with the *non-unitary* $\mathfrak{gl}(1|1)$ representation $\mathscr{V}_\lambda$, whereas the super oscillators are identified with bosonic and fermionic canonical oscillators in accordance with the rule

$$\overline{\xi}^{13} \to \mathbf{c}^\dagger, \quad \xi^{31} \to \mathbf{c}, \quad \overline{\xi}^{23} \to \mathbf{b}^\dagger, \quad \xi^{32} \to \mathbf{b}. \tag{75}$$

The solution to Eq. (23) is then given in the form

$$\mathbb{L}_j^\omega(z) = \mathbf{L}_j^{[1]\,\omega}(z)\mathbf{L}_{-j}^{[2]\,\overline{\omega}}(-z), \tag{76}$$

where $\mathbf{L}_j^\omega(z)$ now implements the auxiliary algebra $\mathfrak{A}_{1,2}^{\{1,2\}} = \mathfrak{gl}(1|1) \otimes \mathfrak{osc}(1|1)$, and the representation parameters take the values

$$z = \frac{1}{2}, \qquad j = \frac{1}{2} + \frac{i}{g}, \qquad \omega = (-|-,+). \tag{77}$$

In particular, the signature labels $\omega$ (where the bar in Eq. (76) stands for flipping the signs) indicate that the auxiliary algebra $\mathfrak{A}_{1,2}^{\{1,2\}}$ should be realized in $\mathscr{V}_\lambda^- \otimes \mathscr{F}^- \otimes \mathscr{B}^+$.

The same procedure can be repeated for the case of bosonic driving (c), where the jump operators act as

$$A_1 = \sqrt{g}\, c_{1,\downarrow}^\dagger c_{1,\uparrow}, \qquad A_N = \sqrt{g}\, c_{i,\uparrow}^\dagger c_{i,\downarrow}. \tag{78}$$

The auxiliary algebra $\mathfrak{A}_{1,2}^{\{2,3\}} = \mathfrak{sl}(2) \otimes \mathfrak{osc}(0|2)$ now consists of a non-compact $\mathfrak{sl}(2)$ module $\mathscr{V}_j$ (with the spin generators denoted by $\mathbf{J}^a$) and a pair of fermionic Fock spaces $\mathscr{F} \otimes \mathscr{F}$,

$$\mathbf{L}_{\{2,3\}}(z) = \begin{pmatrix} 1 & \mathbf{c}_1 & \mathbf{c}_2 \\ \mathbf{c}_1^\dagger & z + \mathbf{J}^3 + \mathbf{c}_1^\dagger \mathbf{c}_1 - \frac{1}{2} & \mathbf{J}^- + \mathbf{c}_1^\dagger \mathbf{c}_2 \\ \mathbf{c}_2^\dagger & \mathbf{J}^+ + \mathbf{c}_2^\dagger \mathbf{c}_1 & z + \mathbf{J}^3 + \mathbf{c}_2^\dagger \mathbf{c}_2 - \frac{1}{2} \end{pmatrix}. \tag{79}$$

In order to fulfil the boundary constraints, the auxiliary algebra of $\mathbf{L}_j^\omega(z)$ should consist of the product $\mathscr{V}_j^- \otimes \mathscr{F}^- \otimes \mathscr{F}^+$. Finally, $\rho_\infty$ is cast in the universal form Eq. (21), where now

$$\mathbb{L}_j^\omega(z) = \mathbf{L}_j^{[1]\,\omega}(z)\mathbf{L}_{-j}^{[2]\,\overline{\omega}}(-z) \qquad \text{with} \qquad z = -\frac{1}{2}, \quad j = \frac{i}{g}, \quad \omega = (-|-,+). \tag{80}$$

# 6  Vacuum Q-operators

Given that all the solutions are directly related to a particular type of solutions of the graded Yang–Baxter equation (8), it is quite remarkable (and perhaps surprising) that a one-parametric family of density matrices $\rho_\infty(g)$ *do not* commute for different values of $g$ (reported first in [117])

$$\left[\rho_\infty(g), \rho_\infty(g')\right] \neq 0, \qquad \text{for} \quad g \neq g'. \tag{81}$$

One may still wish to argue that due to amplitude factorization (18) the nonequilibrium density operators $\rho_\infty(g)$ are not the most 'fundamental' physical objects. Indeed, amplitudes operators $\Omega_N(g)$ themselves *do commute* for different values of couplings,

$$\left[\Omega_N(g), \Omega_N(g')\right] = 0. \tag{82}$$

This shows that $\Omega_N(g)$ can be regarded as a family of *vacuum highest-weight transfer matrices*. However, while the steady states $\rho_\infty(g)$ are diagonalizable objects which encode physical properties of the system, their $\Omega$-amplitudes exhibit a non-trivial *Jordan structure*.[18] Below we examine this unusual behaviour in more detail and relate it to the vacuum Q-operators.

---

[18]This is somewhat reminiscent to what occurs in logarithmic conformal field theories which are governed by non-unitary representations of Virasoro algebra and possess non-diagonalizable dilatation generators.

**Baxter's Q-operators.** Before introducing the notion of vacuum Q-operators, let us first make some comments on the connection between the conventional Q-operators and Lax operators $\mathbf{L}_I(z)$ introduced in Section 4. The concept of a Q-operator was originally introduced in Baxter's seminal paper on the 8-vertex model, where it was used as a device to diagonalize the transfer matrix of the problem by solving a suitable second-order difference relation [118], and later revived in the context of Potts model [119] and integrable structure of CFTs [120, 121]. For clarity we focus the subsequent discussion entirely on the homogeneous $\mathfrak{su}(2)$ spin chain, providing only a condensed summary of the main ingredients. For a more comprehensive and pedagogical exposition we refer the reader to [85].

Baxter's TQ-relation is a functional relation for the fundamental transfer operator $T_\square$ of the form[19]

$$T_\square(z)Q_\pm(z) = T_0(z - \tfrac{1}{2})Q_\pm(z+1) + T_0(z + \tfrac{1}{2})Q_\pm(z-1), \quad T_0(z) = z^N. \tag{83}$$

The pair of Baxter Q-operators $Q_\pm(z)$ represents two *independent* operator solutions to the functional equation (83) which enjoy the involution property

$$\left[T_\square(z), Q_\pm(z')\right] = \left[Q_+(z), Q_-(z')\right] = 0, \qquad \forall z, z' \in \mathbb{C}. \tag{84}$$

Eigenvalues of $Q_\pm(z)$, denoted by $\mathscr{Q}_\pm(z)$, are (up to a twist-dependent phase which is omitted for brevity) *polynomials* of the form

$$\mathscr{Q}_-(z) = \prod_{k=1}^{M}(z - z_k), \qquad \mathscr{Q}_+(z) = \prod_{k=1}^{N-M}(z - \tilde{z}_k). \tag{85}$$

Their zeros $z_k$ ($\tilde{z}_k$) coincide with the Bethe (dual) roots, and are solutions to the celebrated Bethe quantization condition

$$\left(\frac{z+1/2}{z-1/2}\right)^N e^{\pm i\phi} = -\frac{\mathscr{Q}_\mp(z+1)}{\mathscr{Q}_\mp(z+1)}. \tag{86}$$

Polynomiality of eigenvalues of $T_\square(z)$ and $Q_\pm(z)$ ensures that the TQ-relation (83) is *equivalent* to Bethe equations (86).

Operators $Q_\pm$ can be conveniently cast as auxiliary traces over quantum monodromies obtained by the lattice path integration of partonic Lax operators. Specifically, in the $\mathfrak{su}(2)$ case we have

$$Q_\pm(z) \simeq \mathrm{Tr}_{\mathscr{F}}\left(e^{-i\phi\,\mathbf{n}}\mathbf{L}_\pm(z) \otimes \cdots \otimes \mathbf{L}_\pm(z)\right). \tag{87}$$

Here we have made identifications $\mathbf{L}_{\{1\}}(z) \equiv \mathbf{L}_+(z)$, $\mathbf{L}_{\{2\}}(z) \equiv \mathbf{L}_-(z)$, and the trace is with respect to the auxiliary Fock space $\mathscr{F}$. An analogous construction applies to integrable theories based on higher-rank algebras [120–123] where a *complete* set of Q-operators is associated to Lax operators $\mathbf{L}_I(z)$ introduced in Section 4. In the language of Bethe ansatz, this means that eigenvalues of all Q-operators belonging to rational solutions of Yang–Baxter algebra (cf. Eq. (8)) are polynomials whose roots coincide with Bethe roots belonging to different nesting levels. An explicit construction of the full hierarchy of Q-operators for $\mathfrak{gl}(n|m)$ spin chains can be found in [85–88] (and in [124, 125], using a different approach). The outcome of this procedure is a set of $2^{n+m}$ distinct Q-operators which can be arranged on vertices of a hypercubic lattice [126]. In this context, partonic Lax operators $\mathbf{L}_{\{a\}}(z)$ are associated to the distingusihed set of $n + m$ elementary Q-operators $Q_{\{a\}}(z)$ which can be used to solve the spectral problem by explicit integration of an auxiliary linear problem [122,123]. An important consequence of this is that eigenvalues of fused transfer matrices which obey the T-system functional identities [127, 128] decompose in terms of the elementary Q-functions.

---

[19]For technical reasons we shall think of a closed system and impose twisted boundary conditions. The case of periodic boundary conditions (i.e. the limit of vanishing twist $\phi \to 0$) exhibits a subtle singular behaviour due to restoration of the $SU(2)$ multiplets and has to be treated with care (see [85]).

**Vacuum Q-operators.** Since in *open* quantum spin chains translational symmetry is manifestly absent, taking (super) traces over auxiliary spaces is no longer a priori justified. As originally noticed in [72], one may instead use projections onto the highest (or lowest) weight states of auxiliary spaces.[20] To this end we now define the following set of 'vacuum Q-operators'

$$Q_I^{\text{vac}}(z) = \langle \text{vac} | \mathbf{L}_I(z) \circledast \cdots \circledast \mathbf{L}_I(z) | \text{vac} \rangle. \tag{88}$$

The previous analysis of the steady-state solutions for $\mathfrak{gl}(n|m)$ spin chains with integrable dissipative boundaries given by Eq. (24) shows that all $\Omega$-amplitudes can indeed be identified with vacuum Q-operators. More specifically, $\Omega$-amplitudes which enter in our nonequilibrium setting always correspond to 'mesonic' Lax operators $\mathbf{L}_I(z)$ with $|I| = 2$.

We wish to elaborate on a subtle (but important) point in regard to the auxiliary algebra of $\mathbf{L}_I(z)$ and the structure of the vacuum state $|\text{vac}\rangle$. The fact that $\mathbf{L}_I(z)$ carry (besides Dynkin labels) the information about the types of irreducible components which enclose the auxiliary algebra becomes crucial here. For instance, already in the simplest case of the $\mathfrak{su}(2)$ chain, we had to define and operate with two *inequivalent* types of vacuum Q-operators (denoted by $Q_{\{a\}}^{\text{vac},\pm}(z)$, with $a = 1, 2$) associated to the two inequivalent bosonic Fock spaces $\mathscr{B}^\pm$. The explicit structure of the fusion relation for partonic operators $\mathbf{L}_{\{a\}}(z)$ (see appendix B) brings us to the conclusion that the vacuum Q-operators with equal auxiliary modules are still in involution

$$\left[ Q_{\{a\}}^{\text{vac},\pm}(z), Q_{\{a'\}}^{\text{vac},\pm}(z') \right] = 0, \qquad \forall z, z' \in \mathbb{C}, \quad \text{and} \quad a, a' \in \{1, 2\}, \tag{89}$$

which in turn implies that the same property also holds for the corresponding $\Omega$-amplitudes (as given by Eq. (82)). Conversely, the objects which involve inequivalent auxiliary spaces *do not commute*,

$$\left[ Q_{\{a\}}^{\text{vac},\pm}(z), Q_{\{a'\}}^{\text{vac},\mp}(z') \right] \neq 0, \qquad \forall z, z' \in \mathbb{C}, \quad \text{and} \quad a, a' \in \{1, 2\}. \tag{90}$$

Since the steady-state density operators $\rho_\infty(g)$ *always consist of two fused mesonic vacuum Q-operators of the opposite type* (which is a corollary of property (18)), by virtue of Eq. (90) they do not inherit the involution property (82) from their amplitude operators $\Omega_N(g)$.[21] It remains an interesting open problem to devise a suitable generalization of the Algebraic Bethe Ansatz procedure to diagonalize $\rho_\infty$ [117].

# 7 Conclusion and outlook

In this work we introduced a unifying algebraic description of exact nonequilibrium steady states which belong to an important class of integrable quantum lattice models. We presented an explicit construction of density matrices which appear as non-trivial stationary solutions to a non-unitary relaxation process in which a system is coupled to effective Markovian particle reservoirs attached at its boundaries. We employed a simple set of incoherent particle source and drain reservoirs which naturally generalize those used previously in refs. [73, 74, 76, 80, 81]. We have shown that such reservoirs partially preserve the integrable structure of the bulk Hamiltonian and permit to obtain analytic closed-form steady-state density operators in a systematic way.

---

[20]In a more general setting, when particle source and drain terms are rotated with respect to the $z$-axis, the highest-or lowest-weight vacua get replaced by spin-coherent states [129].

[21]It is instructive to remark that tensor products of irreps of mixed types do not admit a resolution in terms of a (finite or infinite) discrete sum over extremal-weight irreps, in contrast to the ubiquitous decomposition of tensor products of finite dimensional irreps (or products of extremal-weight irreps of the same type).

The solutions were presented in the universal form of a homogeneous fermionic matrix-product operators, and shown to decompose in terms of the vacuum analogues of Baxter's Q-operators. Such a factorization property reflects the chiral structure of the states and also allows to reverse directions of particle currents with aid of suitable particle-hole transformations. The basic building blocks of our construction are the so-called partonic Lax operators which stem from certain degenerate representations of graded Yangians, identified recently in [85–88]. These rather unconventional algebraic structures admit a canonical realization in terms spins and oscillators. In the context of our application, these appeared as the auxiliary degrees of freedom in the matrix-product operator representation for the steady-state solutions.

The absence of translational symmetry in open quantum chains is of profound importance and requires to replace the usual auxiliary traces by the projectors onto highest- or lowest-weight auxiliary vacua. The internal algebraic structure of the amplitude operators depends crucially on the parities assigned to the particles which experience dissipation. In the case of equal parities (bosonic driving), the amplitude operators always involve a single auxiliary non-compact $\mathfrak{sl}(2)$ spin, whereas the opposite parities (fermionic driving) require non-unitary irreducible $\mathfrak{gl}(1|1)$ representations which are two dimensional. The residual auxiliary degrees of freedom pertain to a finite number of canonical (bosonic or fermionic) oscillators which remain intact upon varying the coupling parameters of the reservoirs. The universal structure of the steady-state solutions signifies that it is the non-unitary part of the auxiliary algebra which ultimately controls their qualitative characteristics: on one end, the presence of $\mathfrak{sl}(2)$ sectors leads to a universal anomalous (i.e. non-diffusive) $j \sim \mathcal{O}(N^{-2})$ decay of longitudinal currents and cosine-shaped density profiles as already found in [73, 76, 80, 82]. Fermionic driving is on the other hand characterized by $\mathfrak{gl}(1|1)$ subspaces and triggers ballistic transport with non-decaying currents $j \sim \mathcal{O}(N^0)$ and flat density profiles [98]. The solutions at hand can therefore be perceived two particular nonequilibrium universality classes.

The distinguished feature of integrable steady states addressed in this work are the non-unitary representations of Lie (super)algebras . This contrasts the conventional approaches to quantum integrable systems which are primarily based on unitary representations and directly relate to physical excitations in the spectrum (described by the formalism of the Thermodynamic Bethe Ansatz [130–132]). Physical significance of non-unitary representations in integrable theories is on the other hand far less understood and has not been much explored in the literature, although a few prominent examples are worth mentioning. Most notably, the logarithmic CFTs are based on (non-unitary) reducible indecomposable representations of Virasoro algebra [133, 134] and are known to capture various phenomena in statistical physics ranging from critical dense polymers [135], symplectic fermions [136, 137], critical percolation [138–140] to Gaussian disordered systems [141, 142]. It is perhaps instructive to add that non-compact spin chains are also found in the hadron scatting in QCD, which is in the Regge regime governed by the $s = 0$ non-compact isotropic Heisenberg magnet [143, 144].

The role of non-unitarity in the present nonequilibrium setting is however different as it is not (at least directly) attributed to physical degrees of freedom, but instead enters on the level of fictitious particles assigned to auxiliary spaces in a matrix-product representation of nonequilibrium steady states. Nevertheless, it has been found that non-unitary representations can sometimes be linked to certain hidden conservation laws which turn out to be responsible for anomalous quantum spin transport (singular DC conductivity) in the linear-response regime [72, 77, 145, 146].

In the conclusion we wish to highlight a few unresolved aspects of the problem which in our opinion deserve to be better explored and understood. In order to further extend the range of applicability of the present approach, it is of paramount importance to obtain better theoretical understanding of the integrability-preserving dissipative boundaries. In particu-

lar, whether there exist a connection between quantum integrability and a special type of Lindblad reservoirs employed here remains unanswered at the moment. Another intriguing open question is to find a field-theoretic version of the Lindbladian evolution which would qualitatively reproduce the scaling regime of integrable quantum lattices (cf. [80]). It is more-over difficult to overlook several discernible similarities with the Caldeira–Leggett approach of modelling a dissipative environment with a boundary-localized friction term [147,148], which has been applied to sine–Gordon theory with an integrable boundary perturbation [149]. In particular, (i) the boundary current is given by the vacuum eigenvalues of the CFT analogues of Q-operators, (ii) the reservoir parameters are linked to purely imaginary values of high-est weights, and (iii) the particle current is expressed directly in terms of the nonequilibrium partition function $Z = \text{Tr}\,\varrho_\infty$, which is otherwise common to both the asymmetric classical ex-clusion processes [60] and their quantum counterparts [80] considered here. In our opinion, these curiosities deserve to be further explored in future studies.

## Acknowledgements

The author thanks P. Claeys, V. Popkov, E. Quinn, and especially T. Prosen for valuable and stimulating discussions and/or providing comments on the manuscript.

**Funding information.** The author acknowledges support by VENI grant by the Netherlands Organisation for Scientific Research (NWO).

## A  Graded vector spaces and Lie superalgebras

A graded vectors space is a complex vector space $\mathbb{C}^{n|m}$, spanned by basis states $\{|a\rangle\}_{a=1}^{n+m}$, which is endowed with a $\mathbb{Z}_2$ map,

$$p: \qquad \{1,2,\ldots,n+m\} \to \{0,1\}, \tag{91}$$

referred to as the *grading*:

$$p(a) \equiv |a| = \begin{cases} 0 & \text{if} \quad a \text{ is bosonic} \\ 1 & \text{if} \quad a \text{ is fermionic} \end{cases}. \tag{92}$$

We subsequently adopt (with no loss of generality) the *distinguished* grading,

$$|a| = \begin{cases} 0 & a \in \{1,2,\ldots,n\} \\ 1 & a \in \{n+1,\ldots,n+m\} \end{cases}. \tag{93}$$

This assignment induces a block decomposition on $\text{End}(\mathbb{C}^{n|m})$, being the space of matrices acting on $\mathbb{C}^{n|m}$. Specifically, $\text{End}(\mathbb{C}^{n|m}) = \mathscr{V}_0 \oplus \mathscr{V}_1$, where components $\mathscr{V}_0$ (dim $\mathscr{V}_0 = n$) and $\mathscr{V}_1$ (dim $\mathscr{V}_1 = m$) represent bosonic (even) and fermionic (odd) parts, respectively. The subspaces $\mathscr{V}_0$ and $\mathscr{V}_1$ are referred to as the homogeneous components of $\text{End}(\mathbb{C}^{n|m})$. Notice that while $\mathscr{V}_0$ is a subalgebra, the odd part $\mathscr{V}_1$ is not. A vector space $\text{End}(\mathbb{C}^{n|m})$ also constitutes $\mathfrak{gl}(n|m)$ Lie superalgebra. In particular, any element $A$ admits a block form

$$A = \begin{pmatrix} A_{00} & A_{01} \\ A_{10} & A_{11} \end{pmatrix}, \tag{94}$$

where sub-matrices $A_{00}, A_{11}, A_{01}$ and $A_{10}$ are of dimensions $n \times n$, $m \times m$, $n \times m$ and $m \times n$, respectively. The bosonic part decomposes in terms of bosonic subalgebras $\mathfrak{gl}(n|m)_0 \cong \mathfrak{gl}(n) \oplus \mathfrak{gl}(m)$ and corresponds to $A_{01} = A_{10} \equiv 0$, whereas the fermionic (odd) part $\mathfrak{gl}(n|m)_1$ pertains to elements with $A_{00} = A_{11} \equiv 0$.

Let $E^{ab}$ denote matrix units, i.e. matrices with the only non-zero element being 1 in the $a$-th row and $b$-th column. Basis elements $E^{ab}$ are assigned a $\mathbb{Z}_2$-parity according to $p(E^{ab}) \equiv |E^{ab}| \to \{0, 1\}$, with the prescription

$$|E^{ab}| = |a| + |b|. \tag{95}$$

Element $A$ is of even (odd) parity when non-vanishing blocks $A_{ab}$ are of equal (opposite) parity $|a| + |b| = 0$ ($|a| + |b| = 1$). For instance, $R$-matrices $R^{n|m}$ are always even elements. Matrix units $E^{ab}$ form a basis of the fundamental representation of $\mathfrak{gl}(n|m)$ algebra denoted by $\mathcal{V}_{\square}^{n|m}$. The graded Lie bracket is prescribed by

$$\left[A, B\right] := AB - (-1)^{AB} BA, \tag{96}$$

and the graded Jacobi identity reads

$$(-1)^{AC}\left[A, \left[B, C\right]\right] + (-1)^{AB}\left[B, \left[C, A\right]\right] + (-1)^{BC}\left[C, \left[A, B\right]\right] = 0. \tag{97}$$

Here and below we simplified the notation by writing $(-1)^a$ instead of $(-1)^{|a|}$.

Graded vector spaces and Lie superalgebras are naturally extended over $N$-fold product spaces. Product spaces inherit the parity according to the prescription

$$|E^{a_1 b_1} \otimes E^{a_2 b_2} \otimes \cdots \otimes E^{a_N b_N}| = \sum_{k=1}^{N} \left(|a_k| + |b_k|\right). \tag{98}$$

A linear operator $A$ on $(\mathbb{C}^{n|m})^{\otimes N}$ is called a homogeneous element of parity $|A|$ if it satisfies

$$(-1)^{\sum_k (a_k + b_k)} A_{a_1 \ldots a_N, b_1 \ldots b_N} = (-1)^A A_{a_1 \ldots a_N, b_1 \ldots b_N}. \tag{99}$$

A product of two homogeneous elements $A$ and $B$ has a good parity and is given by $|AB| = |A| + |B|$. The presence of non-trivial grading also affects the tensor product. The graded tensor product is denoted by $\circledast$ and is defined as

$$A \circledast B = (-1)^{|A| + r(B)} A \otimes B, \tag{100}$$

where function $r$ designates the row parity,

$$r(E^{a_1 b_1} \otimes E^{a_2 b_2} \otimes \cdots \otimes E^{a_N b_N}) = \sum_{k=1}^{N} r(a_k). \tag{101}$$

The advantage of definition (100) is that it preserves the standard tensor multiplication rule,

$$(A \circledast B)(C \circledast D) = AC \circledast BD. \tag{102}$$

The graded tensor product can be extended to product spaces by introducing (homogeneous) elements $E_j^{ab}$, representing the generators associated to the $j$-th copy of $\text{End}(\mathbb{C}^{n|m})$. Notice that in contrast to the standard (non-graded) basis $1 \otimes \cdots \otimes E^{ab} \otimes \cdots \otimes 1$ of $\text{End}(\mathbb{C}^{n+m})^{\otimes N}$, elements $E_j^{ab}$ do not commute at different lattice sites, but we find instead

$$E_i^{ab} E_j^{cd} = (-1)^{(a+b)(c+d)} E_j^{cd} E_i^{ab}. \tag{103}$$

On the same lattice site they however still obey the property of projectors,

$$E_i^{ab} E_i^{cd} = \delta_{cb} E_i^{ad}. \tag{104}$$

The last two properties combined yield

$$\left[ E_j^{ab}, E_k^{cd} \right] = \delta_{jk} \left( \delta_{cb} E_k^{ad} - (-1)^{(a+b)(c+d)} \delta_{ad} E_j^{cb} \right). \tag{105}$$

The graded generators acting on the $N$-particle space $(\mathbb{C}^{n|m})^{\otimes N}$ read in terms of the graded tensor product

$$E_i^{ab} = 1^{\otimes(i-1)} \circledast E^{ab} \circledast 1^{\otimes(N-i)}, \tag{106}$$

whereas expressed in terms of the standard tensor product they assume the expansion

$$E_i^{ab} = (-1)^{(a+b)\sum_{k=j+1}^{N} c_k} 1^{\otimes(i-1)} \otimes E^{ab} \otimes E^{c_{j+1}c_{j+1}} \otimes \cdots \otimes E^{c_N c_N}. \tag{107}$$

This prescription should be interpreted as the higher-rank version of the Jordan–Wigner transformation [150].

Interaction densities $h^{n|m}$ for a class of the so-called 'fundamental graded models' are identified with graded permutations $P^{n|m}$ on $\mathbb{C}^{n|m} \otimes \mathbb{C}^{n|m}$,

$$P^{n|m} = (-1)^b E^{ab} \circledast E^{ba}. \tag{108}$$

Permutations $P^{n|m}$ can be alternatively given also as matrices acting on the two-fold fundamental spaces $\mathbb{C}^{n+m} \otimes \mathbb{C}^{n+m}$, reading

$$P^{n|m} = (-1)^{a+b} E^{ab} \otimes E^{ba}. \tag{109}$$

The defining $\mathfrak{su}(n|m)$ representations admit realizations in terms of canonical fermions. In the $\mathfrak{su}(1|1)$ case, the graded projectors $\mathscr{E}_i$ act non-identically only on the $i$-th copy of $\mathbb{C}^{1|1}$ in the chain, and are realized as a $2 \times 2$ matrix of spinless fermions

$$\mathscr{E}_i = \begin{pmatrix} 1 - n_i & c_i \\ c_i^\dagger & n_i \end{pmatrix}. \tag{110}$$

Here the generators $n_i$ and $1 - n_i$ span the even (bosonic) subalgebra $\mathscr{V}_0$, while $c_i$ and $c_i^\dagger$ are the fermionic generators which span the odd part $\mathscr{V}_1$ and satisfy canonical anticommutation relations

$$\{c_i, c_j^\dagger\} = \delta_{j,k}, \quad \{c_i, c_j\} = \{c_i^\dagger, c_j^\dagger\} = 0. \tag{111}$$

Equation (107) is nothing but the well-known Jordan–Wigner transformation from Pauli spins to canonical spinless fermions

$$\begin{aligned} c_i^\dagger &= 1^{\otimes(i-1)} \otimes \sigma^- \otimes (\sigma^z)^{\otimes(N-1)}, \\ c_i &= 1^{\otimes(i-1)} \otimes \sigma^+ \otimes (\sigma^z)^{\otimes(N-1)}. \end{aligned} \tag{112}$$

In systems with multiple fermionic species (e.g. spin-carrying fermions), the super projectors can be constructed with aid of the fusion procedure [150]. For instance, the local physical space of a $\mathfrak{su}(2|2)$ spin chain is four dimensional, spanned by states

$$|0\rangle, \quad |\uparrow\rangle \equiv c_\uparrow^\dagger |0\rangle, \quad |\downarrow\rangle \equiv c_\downarrow^\dagger |0\rangle, \quad |\uparrow\downarrow\rangle \equiv c_\uparrow^\dagger c_\downarrow^\dagger |0\rangle. \tag{113}$$

At each lattice site $i$ we thus have $4 \times 4 = 16$ generators,

$$\mathscr{E}_{\uparrow\downarrow}^{ac,bd} = (\mathscr{E}_\uparrow \circledast \mathscr{E}_\downarrow)^{ac,bd} \equiv (-1)^{|a+b||c|} \mathscr{E}_\uparrow^{ab} \mathscr{E}_\downarrow^{cd}. \tag{114}$$

Flattening the indices readily yields the graded permutation on $\mathbb{C}^{2|2} \otimes \mathbb{C}^{2|2}$, taking the form

$$P^{2|2} = (-1)^b \mathscr{E}^{ab}_{\uparrow\downarrow} \otimes \mathscr{E}^{ba}_{\uparrow\downarrow}. \tag{115}$$

Furthermore, the fermionic realization of the graded projector $P^{2|1}$ can be obtained from $P^{2|2}$ by projecting out e.g. the doubly-occupied state $|\uparrow\downarrow\rangle$. The local space of states thus consists of the triplet

$$|0\rangle, \quad c^\dagger_\uparrow |0\rangle, \quad c^\dagger_\downarrow |0\rangle. \tag{116}$$

Choosing e.g. the grading as $|0| = 0$ and $|1| = |2| = 1$, one finds

$$\mathscr{E}_i = \begin{pmatrix} (1-n_{i,\uparrow})(1-n_{i,\downarrow}) & (1-n_{i,\uparrow})c_{i,\downarrow} & c_{i,\uparrow}(1-n_{i,\downarrow}) \\ (1-n_{i,\uparrow})c^\dagger_{i,\downarrow} & (1-n_{i,\uparrow})n_{i,\downarrow} & c^\dagger_{i,\downarrow}c_{i,\uparrow} \\ c^\dagger_{i,\uparrow}(1-n_{i,\downarrow}) & c^\dagger_{i,\uparrow}c_{i,\downarrow} & n_{i,\uparrow}(1-n_{i,\downarrow}) \end{pmatrix}. \tag{117}$$

# B   Fusion and factorization properties of Lax operators

In this section we revisit the main factorization and fusion formulae for the $\mathfrak{gl}(n|m)$ integrable spin chains. A comprehensive and more detailed presentation can be found e.g. in references [85–88].

**Classification of irreducible highest-weight $\mathfrak{gl}(n|m)$ modules.**   Before presenting the fundamental factorization property of the rational Lax operators we need to introduce irreducible highest-weight $\mathfrak{gl}(n|m)$ representations. These are known as the *Verma modules*, denoted by $\mathscr{V}^+_{\Lambda_{n+m}}$, and are characterized by (i) the highest-weight property

$$\mathbf{J}^{a,a+1} |\text{hws}\rangle = 0 \qquad \text{for} \qquad a = 1, 2, \ldots, n+m-1, \tag{118}$$

and (ii) the weight vector $\Lambda_{n+m} = (\lambda_1, \ldots, \lambda_n, \lambda_{n+1}, \ldots, \lambda_{n+m})$ through the action of Cartan generators (no summation over repeated indices),

$$\mathbf{J}^{aa} |\text{hws}\rangle = \lambda_a |\text{hws}\rangle. \tag{119}$$

A representation $\Lambda_n$ is typically of *infinite* dimension, corresponding to generic complex-valued weights $\lambda_a$. Since we shall mostly need the restriction to $\mathfrak{sl}(n|m)$ subalgebra, we introduce the $\mathfrak{sl}(n|m)$ weights as

$$\mu_a = (-1)^a \lambda_a - (-1)^{a+1}\lambda_{a+1}, \qquad a = 1, 2, \ldots, n-1. \tag{120}$$

In the case of unitary $\mathfrak{sl}(n|m)$ representations, all $\mu_a$ for $a \neq n$ must be non-negative integers, while $\mu_n$ can take arbitrary real values. The fundamental representation of $\mathfrak{sl}(n|m)$ is given by the weight vector $\Lambda_{n+m} = (1, 0, \ldots, 0)$. Kac–Dynkin labels and finite-dimensional irreducible representations are in one-to-one correspondence, with Young diagrams corresponding to non-negative non-increasing weights. Rectangular representations $\{s, a\}$ with $s$ columns and $a$ rows have a single non-vanishing label $\mu_a = s$.

## B.1   Factorization of Lax operators

Highest-weight $\mathfrak{gl}(n|m)$-invariant transfer operators $T^+_{\Lambda_{n+m}}(z)$ acting on a Hilbert space $\mathscr{H} \cong (\mathbb{C}^{n|m})^{\otimes N}$ are given by[22]

$$T^+_{\Lambda_{n+m}}(z) = \text{Str}_{\mathscr{V}^+_{\Lambda_{n+m}}} \underbrace{\mathbf{L}_{\Lambda_{n+m}}(z) \otimes \cdots \otimes \mathbf{L}_{\Lambda_{n+m}}(z)}_{N \text{ copies}}, \tag{121}$$

---

[22]Here it is implicitly assumed that the super trace exists. Additional regulators in the form of boundary twists may be needed in general.

which due to Yang–Baxter relation (8) enjoy the commutativity property,

$$\left[T^+_{\Lambda_{n+m}}(z), T^+_{\Lambda'_{n+m}}(z')\right] = 0. \tag{122}$$

for all $z, z' \in \mathbb{C}$ and representation labels $\Lambda_{n+m}$ and $\Lambda'_{n+m}$. Remarkably however, operators $T^+_{\Lambda_{n+m}}$ do not represent the most elementary objects in the theory. In fact, they factorize[23] into an ordered sequence of Q-operators $Q_{\{a\}}$,[24]

$$T^+_{\Lambda_{n+m}}(z) \simeq Q_{\{1\}}(z_1 + \lambda'_1)Q_{\{2\}}(z_2 + \lambda'_2)\cdots Q_{\{n+m\}}(z_{n+m} + \lambda'_{n+m}). \tag{123}$$

Here parameters $\lambda'_a$ are the 'shifted weights',

$$\Lambda'_{n+m} = \Lambda_{n+m} + \varrho_{n+m}, \qquad \varrho_a = \sum_{b=a+1}^{n+m} \tfrac{1}{2}(-1)^b - \sum_{b=1}^{a-1} \tfrac{1}{2}(-1)^b. \tag{124}$$

In the $\mathfrak{gl}(n)$ case, i.e. when $m = 0$, the shifts arrange in the 'complete $n$-strings', reading $\varrho_n = \frac{1}{2}(n-1, n-3, \ldots 1-n)$, closely resembling the pattern of the string-type solutions to Bethe Ansatz equations. Indeed, factorization property (123) is a direct consequence of the local factorization relation [86]

$$\mathbf{L}_{\{1\}}(z_1 + \lambda'_1)\mathbf{L}_{\{2\}}(z_2 + \lambda'_2)\cdots\mathbf{L}_{\{n+m\}}(z_{n+m} + \lambda'_{n+m}) = \mathbf{S}\,\mathbf{L}^+_{\Lambda_{n+m}}(z)\,\mathbf{K}\mathbf{S}^{-1}, \tag{125}$$

Below we exemplify the factorization procedure on a few concrete instances.

**Basic example: $\mathfrak{sl}(2)$ case.** The factorization property is best illustrated on the $\mathfrak{sl}(2)$ case. The corresponding highest-weight Lax operators read

$$\mathbf{L}^+_{\Lambda_2}(z) = \begin{pmatrix} z + \mathbf{J}^3 & \mathbf{J}^- \\ \mathbf{J}^+ & z - \mathbf{J}^3 \end{pmatrix}, \tag{126}$$

and are characterized by a single Dynkin label $j$ which parametrizes the $\mathfrak{gl}(2)$ weight vector $\Lambda_2 = (j, -j)$. The non-compact spin generators $\mathbf{J}^a$ act on a $\mathfrak{sl}(2)$ module $\mathcal{V}^+_j$, and can be conveniently given in the Holstein–Primakoff form

$$\mathbf{J}^3 = j - \mathbf{b}^\dagger\mathbf{b}, \quad \mathbf{J}^+ = \mathbf{b}, \quad \mathbf{J}^- = \mathbf{b}^\dagger(2j - \mathbf{b}^\dagger\mathbf{b}), \tag{127}$$

where $\mathbf{b}$ and $\mathbf{b}^\dagger$ are the generators of a bosonic oscillator obeying canonical commutation relations

$$\left[\mathbf{b}, \mathbf{b}^\dagger\right] = 1, \quad \left[\mathbf{h}, \mathbf{b}\right] = -\mathbf{b}, \quad \left[\mathbf{h}, \mathbf{b}^\dagger\right] = \mathbf{b}^\dagger, \tag{128}$$

and $\mathbf{h} = \mathbf{b}^\dagger\mathbf{b} + \frac{1}{2}$ is the mode number operator. We furthermore define two types of Fock space representations, denoted by

$$\begin{aligned} \mathscr{B}^+: & \quad \mathbf{b}\,|0\rangle = 0, \quad \mathbf{b}^\dagger\,|k\rangle = |k+1\rangle, \\ \mathscr{B}^-: & \quad \mathbf{b}^\dagger\,|0\rangle = 0, \quad \mathbf{b}\,|k\rangle = |k+1\rangle. \end{aligned} \tag{129}$$

The two are related to each other under the particle-hole transformation $\mathbf{b} \to \mathbf{b}^\dagger$, $\mathbf{b}^\dagger \to \mathbf{b}$ and $\mathbf{h} \to -\mathbf{h}$.

A pair of partonic Lax operators $\mathbf{L}_{\{1\}}(z)$ and $\mathbf{L}_{\{2\}}(z)$ can be straightforwardly obtained from $T^+_{\Lambda_2}(z)$ by considering two possible ways of taking a (correlated) large-$j$ and large-$z$ limits

---

[23]The algebraic origin of the factorization formula has to do with the $\mathscr{U}(\mathfrak{g})$-invariant universal $\mathscr{R}$-matrix decomposing in terms of tensor products of components from the corresponding Borel subalgebras [120, 121].

[24]Geometrically, Q-operators can understood as Plücker coordinates on Grassmannian manifolds [122, 151].

(cf. [85]). This is achieved by keeping either of the combinations $z_{\pm} = z \pm (j + \frac{1}{2})$ fixed, resulting in the 'degenerate' Lax operators of the form

$$\mathbf{L}_{\{1\}}(z) = \begin{pmatrix} z - \mathbf{h} & \mathbf{b}^{\dagger} \\ -\mathbf{b} & 1 \end{pmatrix}, \qquad \mathbf{L}_{\{2\}}(z) = \begin{pmatrix} 1 & \mathbf{b}^{\dagger} \\ -\mathbf{b} & z + \mathbf{h} \end{pmatrix}, \tag{130}$$

which represent two distinct well-defined solutions to the Yang–Baxter equation.

## B.2 Fusion of partonic Lax operators

Quantum groups are endowed with a coproduct, ensuring that the algebraic structure gets preserved under tensor multiplication $\mathscr{Y} \to \mathscr{Y} \circledS \mathscr{Y}$. Partonic Lax operators, as defined in Eq. (42), represent the simplest solutions of the Yang–Baxter equation (8). They serve as irreducible components for obtaining other realizations of $\mathscr{Y}$ via fusion. Below we outline the main features of such a procedure, while referring the reading for a more complete and detailed presentation to references [86–88].

Let $I, J \subseteq \{1, 2, \dots, n + m\}$ be two index sets. We shall only consider operators $\mathbf{L}(z)$ which are linear in spectral parameter $z$, requiring $I$ and $J$ to be non-intersecting, $I \cap J = \emptyset$. Set $I$ (resp. $J$) comprises of $p$ ($\dot{p}$) bosonic and $q$ ($\dot{q}$) fermionic indices. We furthermore introduce $K \equiv I \cup J$, involving $\ddot{p}$ ($\ddot{q}$) bosonic (fermionic) indices, such that $p + \dot{p} + \ddot{p} = n$ and $q + \dot{q} + \ddot{q} = m$. Fusion is a process of merging two canonical Lax operators $\mathbf{L}_I$ and $\mathbf{L}_J$ of respective ranks $|I| = p + q$ and $|J| = \dot{p} + \dot{q}$, which takes the abstract form

$$\mathbf{L}_K(z) \sim \mathbf{L}_I^{[1]}(z + z_1) \mathbf{L}_J^{[2]}(z + z_2), \tag{131}$$

for some appropriate choice of shifts $z_1$ and $z_2$. The superscript square brackets were needed here to distinguish inequivalent species. The precise prescription for the fusion rule is entailed by the following form

$$\mathbf{L}_I^{[1]}\Big(z + \tfrac{1}{2} \sum_{\dot{d} \in J} (-1)^{\dot{d}}\Big) \mathbf{L}_J^{[2]}\Big(z - \lambda - \tfrac{1}{2} \sum_{d \in I} (-1)^d\Big) = \mathbf{S} \, \mathbf{L}_K^{[1]}(z) \, \mathbf{K}^{[2]} \, \mathbf{S}^{-1}, \tag{132}$$

where $\mathbf{K}$ is a triangular 'disentangling matrix' and $\mathbf{S}$ a suitable global similarity transformation. An implication of fusion formula (132) is that Lax operators which are realized in terms of algebras $\mathfrak{A}_{m,n}^K$ are not elementary, but instead factorize according to $\mathfrak{A}_{m,n}^K \to \mathfrak{A}_{m,n}^I \otimes \mathfrak{A}_{m,n}^J$. Below we take a closer look at this by inspecting a few explicit examples.

We begin by noticing that the above fusion procedure clearly violates the canonical form given by Eq. (40), as it appears to involve an exceeding number of auxiliary spaces. We shall in turn demonstrate that all redundant auxiliary spaces can be eliminated upon appropriately redefining the generators. In particular, there exist a canonical procedure to reduce the number of oscillators from $|I| \cdot |\bar{I}| + |J| \cdot |\bar{J}|$ down to $|K| \cdot |\bar{K}|$ and expressing the $\mathfrak{gl}(|K|)$ generators in terms of independent generators of $\mathfrak{gl}(|I|)$ and $\mathfrak{gl}(|J|)$, dressed with $|K|$ additional oscillators. This is in practice achieved by virtue of the homomorphisms [87]

$$\mathfrak{gl}(|K|) \to \mathfrak{gl}(p|q) \otimes \mathfrak{gl}(\dot{p}|\dot{q}) \otimes \mathfrak{osc}(p + \dot{p}|q + \dot{q}), \tag{133}$$

in terms of which the post-fusion $\mathfrak{gl}(p + \dot{p}|q + \dot{q})$ generators $\widehat{\mathbf{J}}^{ab}$ are given by the following prescription

$$\begin{aligned}
\widehat{\mathbf{J}}^{ab} &= \mathbf{J}_1^{ab} + \overline{\xi}_1^{a\dot{c}} \xi_1^{\dot{c}b}, \\
\widehat{\mathbf{J}}^{ab} &= \lambda(-1)^b \overline{\xi}_1^{a\dot{b}} - (-1)^{(b+\dot{d})(b+c)} \overline{\xi}_1^{a\dot{d}} \overline{\xi}_1^{c\dot{b}} \xi_1^{\dot{d}c} + \overline{\xi}_1^{a\dot{c}} \mathbf{J}_2^{\dot{c}b} - (-1)^{b+c} \mathbf{J}_1^{ac} \overline{\xi}_1^{c\dot{b}}, \\
\widehat{\mathbf{J}}^{\dot{a}b} &= \xi_1^{\dot{a}b}, \\
\widehat{\mathbf{J}}^{\dot{a}b} &= \mathbf{J}_2^{\dot{a}b} + \lambda(-1)^b \delta_{\dot{a}b} - (-1)^{(\dot{a}+b)(b+c)} \overline{\xi}_1^{c\dot{b}} \xi_1^{\dot{a}c},
\end{aligned} \tag{134}$$

with $\mathbf{J}_1^{ab}$ and $\mathbf{J}_2^{\dot{a}\dot{b}}$ denoting the $\mathfrak{gl}(p|q)$ and $\mathfrak{gl}(\dot{p}|\dot{q})$ super spins, respectively, whereas the oscillators are to be identified as

$$\xi_{\dot{a}b} = \begin{cases} \xi_{\dot{a}b}^{[1]}, & b \in I \\ \xi_{\dot{a}b}^{[2]}, & b \in J \end{cases}, \qquad \overline{\xi}_{a\dot{b}} = \begin{cases} \overline{\xi}_{a\dot{b}}^{[1]}, & a \in I \\ \overline{\xi}_{a\dot{b}}^{[2]}, & a \in J \end{cases}. \tag{135}$$

The latter are either bosonic or fermionic, depending on the grading. Finally, the tridiagonal matrix $\mathbf{K}$ is of the form

$$\mathbf{K} = \begin{pmatrix} 1 & -(-1)^b \xi_2^{a\dot{b}} & 0 \\ 0 & 1 & 0 \\ 0 & 0 & 1 \end{pmatrix}, \tag{136}$$

while the similarity transformation in Eq. (132) reads

$$\mathbf{S} = \exp\left( \sum_{a \in I} \sum_{\dot{b} \in J} \sum_{\ddot{c} \in \overline{K}} \overline{\xi}_1^{a\dot{b}} \left( (-1)^a \overline{\xi}_2^{\ddot{b}a} + \overline{\xi}_2^{\dot{b}\ddot{c}} \xi_1^{\ddot{c}a} \right) \right), \tag{137}$$

where the double-dotted indices represent the summation over $K$.

$\mathfrak{sl}(2)$ **case.** The basic principle of fusion can be explained on the $\mathfrak{sl}(2)$ theory. Fusion can be understood as the opposite procedure of factorization which is outlined in the previous section. The partonic pieces given by expressions (130) can be fused in two distinct ways. To this end we introduce square brackets and assign a bosonic oscillator to each tensor factor, yielding the following operator identity on $\mathbb{C}^2 \otimes \mathscr{B} \otimes \mathscr{B}$,

$$\begin{aligned} \mathbf{L}_{\{2\}}^{[1]}(z_2) \mathbf{L}_{\{1\}}^{[2]}(z_1) &= \begin{pmatrix} 1 & \mathbf{b}_1^\dagger \\ \mathbf{b}_1 & z_2 + \mathbf{n}_1 \end{pmatrix} \begin{pmatrix} z_1 - \mathbf{n}_2 & \mathbf{b}_2^\dagger \\ -\mathbf{b}_2 & 1 \end{pmatrix} \\ &= \exp\left( \mathbf{b}_1^\dagger \mathbf{b}_2 \right) \begin{pmatrix} 1 & 0 \\ \mathbf{b}_1 & 1 \end{pmatrix} \begin{pmatrix} z + \mathbf{J}_2^3 & \mathbf{J}_2^- \\ \mathbf{J}_2^+ & z - \mathbf{J}_2^3 \end{pmatrix} \exp\left( -\mathbf{b}_1^\dagger \mathbf{b}_2 \right), \end{aligned} \tag{138}$$

where the input spectral parameters are given by

$$z_1 = z + j + \tfrac{1}{2}, \qquad z_2 = z - j - \tfrac{1}{2}. \tag{139}$$

Notice that in the second line of Eq. (138) the oscillators have been rearranged using the similarity transformation $\mathbf{S} = \exp\left( \mathbf{b}_1^\dagger \mathbf{b}_2 \right)$ on $\mathscr{B} \otimes \mathscr{B}$, which reads explicitly

$$\mathbf{S} \mathbf{b}_1^\dagger \mathbf{S}^{-1} = \mathbf{b}_1^\dagger, \quad \mathbf{S} \mathbf{b}_1 \mathbf{S}^{-1} = \mathbf{b}_1 + \mathbf{b}_2, \quad \mathbf{S} \mathbf{b}_2^\dagger \mathbf{S}^{-1} = \mathbf{b}_1^\dagger + \mathbf{b}_2^\dagger, \quad \mathbf{S} \mathbf{b}_2 \mathbf{S}^{-1} = \mathbf{b}_2. \tag{140}$$

On the other hand, fusing in the opposite order yields a similar operator identity

$$\mathbf{L}_{\{1\}}^{[1]}(z_+) \mathbf{L}_{\{2\}}^{[2]}(z_-) = \exp\left( \mathbf{b}_1^\dagger \mathbf{b}_2^\dagger \right) \begin{pmatrix} z + \mathbf{J}_1^3 & \mathbf{J}_1^- \\ \mathbf{J}_1^+ & z - \mathbf{J}_1^3 \end{pmatrix} \begin{pmatrix} 1 & -\mathbf{b}_2 \\ 0 & 1 \end{pmatrix} \exp\left( -\mathbf{b}_1^\dagger \mathbf{b}_2^\dagger \right), \tag{141}$$

where again the parameter constraints (139) are imposed.

Formula (138) readily implies the factorization property for the highest-weight $\mathfrak{sl}(2)$-invariant transfer operator $T_{\Lambda_2}^+(z) \equiv T_j^+(z)$ in terms of a pair of Q-operators,

$$T_j^+(z) = Q_{\{1\}}(z + j + \tfrac{1}{2}) Q_{\{2\}}(z - j - \tfrac{1}{2}). \tag{142}$$

A sequence of transfer matrices $T_j(z)$ with $2j \in \mathbb{Z}$, pertaining to finite-dimensional irreducible $\mathfrak{su}(2)$ representations, can the be obtained from $T_j^+(z)$ with aid of the Bernstein–Gelfand–Gelfand resolution of finite-dimensional modules, $\mathscr{V}_j = \mathscr{V}_j^+ - \mathscr{V}_{-j-1}^+$, resulting in Bazhanov–Reshetikhin determinant representation [152]

$$T_j(z) = Q_{\{1\}}(z + j + \tfrac{1}{2}) Q_{\{2\}}(z - j - \tfrac{1}{2}) - Q_{\{2\}}(z + j + \tfrac{1}{2}) Q_{\{1\}}(z - j - \tfrac{1}{2}). \tag{143}$$

A comment in regard to the so-called vacuum Q-operators is in order here. First, recall that the vacuum Q-operators represent a family of transfer operators which are constructed from a path-ordered product of (partonic) Lax operators contracted with respect to a suitable 'vacuum state'. These vacuum states are presently identified with the highest (or lowest) weight state in $\mathscr{B}$. Mutual commutativity of the vacuum Q-operators can be inferred from fusion formula (138), after observing that (i) when the two Fock space involved are of same type the product vacuum $|0\rangle \otimes |0\rangle$ remains inert under the action of the similarity transformation $\mathbf{S}$, i.e. $\mathbf{S} |0\rangle \otimes |0\rangle = |0\rangle \otimes |0\rangle$, and (ii) the disentangler $\mathbf{K}$ has no global effect due to its triangular form. In the opposite scenario, a fusion of two Lax operators which involve two different types of Fock spaces inevitably excites the vacuum to a coherent state which in turn prevents the vacuum T-operator from decomposing into two vacuum Q-operators. For the very same reason the vacuum Q-operators pertaining to inequivalent auxiliary modules are not guaranteed to satisfy the involution property. In fact, it can be explicitly checked that they do not commute.

$\mathfrak{sl}(3)$ **case.** The $\Omega$-amplitudes for the integrable steady states constructed in this work are all formed from the 'mesonic' Lax operators, namely objects which result from the fusion of two partonic elements. As an explicit example we consider the $SU(3)$-symmetric Lai–Sutherland chain [107, 108], to which we ascribe auxiliary algebra $\mathfrak{A}_{3,0}^{\{1\}} \otimes \mathfrak{A}_{3,0}^{\{2\}} \to \mathfrak{A}_{3,0}^{\{1,2\}}$. Setting $I = \{1\}$ and $J = \{2\}$, the fusion formula is of the form (using the shifted weights $\varrho_2 = \frac{1}{2}(1,-1)$)

$$\mathbf{L}_{\{1\}}^{[1]}(z + \lambda + \tfrac{1}{2}) \mathbf{L}_{\{2\}}^{[2]}(z - \tfrac{1}{2}) = \mathbf{S}\, \mathbf{L}_{\{1,2\}}^{[1]}(z)\, \mathbf{K}^{[2]}\, \mathbf{S}^{-1}, \tag{144}$$

with the partonic Lax operators reading

$$\mathbf{L}_{\{1\}}^{[1]}(z) = \begin{pmatrix} z + j_1 - \mathbf{h}_1^{[1]} - \mathbf{h}_2^{[1]} & \mathbf{b}_1^{[1]\dagger} & \mathbf{b}_2^{[1]\dagger} \\ -\mathbf{b}_1^{[1]} & 1 & 0 \\ -\mathbf{b}_2^{[1]} & 0 & 1 \end{pmatrix}, \tag{145}$$

$$\mathbf{L}_{\{2\}}^{[2]}(z) = \begin{pmatrix} 1 & -\mathbf{b}_1^{[2]} & 0 \\ \mathbf{b}_1^{[2]\dagger} & z + j_2 - \mathbf{h}_1^{[2]} - \mathbf{h}_2^{[2]} & \mathbf{b}_2^{[2]\dagger} \\ 0 & -\mathbf{b}_2^{[2]} & 1 \end{pmatrix}. \tag{146}$$

The oscillators are disentangled with aid of

$$\mathbf{K}^{[2]} = \begin{pmatrix} 1 & -\mathbf{b}_1^{[2]} & 0 \\ 0 & 1 & 0 \\ 0 & 0 & 1 \end{pmatrix}, \tag{147}$$

and an additional similarity transformation $\mathbf{S} = \mathbf{U}\mathbf{V}$, with

$$\mathbf{U} = \exp\left(\mathbf{b}_1^{[1]\dagger} \mathbf{b}_1^{[2]\dagger}\right), \quad \mathbf{V} = \exp\left(\mathbf{b}_1^{[1]\dagger} \mathbf{b}_2^{[2]\dagger} \mathbf{b}_2^{[1]}\right). \tag{148}$$

Explicitly, these transformations act as

$$\begin{aligned} \mathbf{U}\mathbf{b}_1^{[1]} \mathbf{U}^{-1} &= \mathbf{b}_1^{[1]} - \mathbf{b}_1^{[2]\dagger}, \\ \mathbf{U}\mathbf{b}_1^{[2]} \mathbf{U}^{-1} &= \mathbf{b}_1^{[2]} - \mathbf{b}_1^{[1]\dagger}, \end{aligned} \tag{149}$$

and

$$\begin{aligned} \mathbf{V}\mathbf{b}_1^{[1]} \mathbf{V}^{-1} &= \mathbf{b}_1^{[1]} - \mathbf{b}_2^{[2]\dagger} \mathbf{b}_2^{[1]}, \\ \mathbf{V}\mathbf{b}_2^{[2]} \mathbf{V}^{-1} &= \mathbf{b}_2^{[2]} - \mathbf{b}_1^{[1]\dagger} \mathbf{b}_2^{[1]}, \\ \mathbf{V}\mathbf{b}_2^{[1]\dagger} \mathbf{V}^{-1} &= \mathbf{b}_2^{[1]\dagger} + \mathbf{b}_1^{[1]\dagger} \mathbf{b}_2^{[2]\dagger}, \end{aligned} \tag{150}$$

respectively. Putting everything together, relabelling the oscillators,

$$\mathbf{b}_2^{[1]} \to \mathbf{b}_1^{[1]}, \quad \mathbf{b}_1^{[2]} \to \mathbf{b}_1^{[1]}, \quad \mathbf{b}_2^{[1]\dagger} \to \mathbf{b}_1^{[1]\dagger}, \quad \mathbf{b}_2^{[2]\dagger} \to \mathbf{b}_2^{[1]\dagger}, \tag{151}$$

and representing the generators of the $\mathfrak{sl}(2)$ spins acting on $\mathcal{V}_j$ (where $2j = j_1 - j_2 + \lambda$) as

$$
\begin{aligned}
\mathbf{J}^{11} &\leftarrow j_1 + \lambda - \mathbf{b}_1^{[1]\dagger}\mathbf{b}_1^{[1]}, \\
\mathbf{J}^{21} &\leftarrow \mathbf{b}_1^{[1]\dagger}\mathbf{b}_1^{[1]\dagger}\mathbf{b}_1^{[1]} - (j_1 - j_2 - \lambda)\mathbf{b}_1^{[1]\dagger}, \\
\mathbf{J}^{12} &\leftarrow -\mathbf{b}^{[1]}, \\
\mathbf{J}^{22} &\leftarrow j_2 - \mathbf{b}_1^{[1]\dagger}\mathbf{b}_1^{[1]},
\end{aligned}
\tag{152}
$$

yields precisely the anticipated canonical representation of the mesonic Lax operator

$$
\mathbf{L}_{\{1,2\}}(z) = \begin{pmatrix} z + \mathbf{J}^{11} - \mathbf{h}_1 & \mathbf{J}^{21} - \mathbf{b}_1^\dagger\mathbf{b}_2 & \mathbf{b}_1^\dagger \\ \mathbf{J}^{12} - \mathbf{b}_2^\dagger\mathbf{b}_1 & z + \mathbf{J}^{22} - \mathbf{h}_2 & \mathbf{b}_2^\dagger \\ -\mathbf{b}_1 & -\mathbf{b}_2 & 1 \end{pmatrix}. \tag{153}
$$

$\mathfrak{gl}(1|1)$ **case.** It may be instructive to also explicitly spell out the fusion step (132) for the $\mathfrak{gl}(1|1)$ Lie superalgebra. The latter is spanned by four elements, two bosonic generator $N$ and $E$, and two fermionic ones $\psi^\pm$. Commutation relations read

$$[N, \psi^\pm] = \pm\psi^\pm, \quad [\psi^-, \psi^+] = E, \quad (\psi^-)^2 = (\psi^+)^2 = 0, \tag{154}$$

where $E$ is the central element. The defining (fundamental) representation is of dimension 2, and is given by $2 \times 2$ matrices

$$E = \begin{pmatrix} 1 & 0 \\ 0 & 1 \end{pmatrix}, \quad N = \begin{pmatrix} 0 & 0 \\ 0 & 1 \end{pmatrix}, \quad \psi = \begin{pmatrix} 0 & 1 \\ 0 & 0 \end{pmatrix}, \quad \psi^\dagger = \begin{pmatrix} 0 & 0 \\ 1 & 0 \end{pmatrix}. \tag{155}$$

By setting $E = 0$, we obtain a trivial one-dimensional irreducible representation with $\psi^\pm \equiv 0$. There moreover exists a family of two-dimensional irreducible representations, denoted by $\langle n, e \rangle$ (with $E \neq 0$), which reads

$$E = \begin{pmatrix} e & 0 \\ 0 & e \end{pmatrix}, \quad N = \begin{pmatrix} n-1 & 0 \\ 0 & n \end{pmatrix}, \quad \psi = \begin{pmatrix} 0 & 1 \\ 0 & 0 \end{pmatrix}, \quad \psi^\dagger = \begin{pmatrix} 0 & 0 \\ e & 0 \end{pmatrix}, \tag{156}$$

and includes the fundamental representation (155) as $\langle 1, 1 \rangle$. Reducible indecomposable representations of $\mathfrak{gl}(1|1)$ (see e.g. [133, 153]) and not of our interest here.

Fusion in the fermionic case works as follows. The partonic Lax operators contain a single fermionic specie and are of the form

$$\mathbf{L}_{\{1\}}(z) = \begin{pmatrix} z - (\mathbf{n} - \frac{1}{2}) & \mathbf{c}^\dagger \\ -\mathbf{c} & 1 \end{pmatrix}, \qquad \mathbf{L}_{\{2\}}(z) = \begin{pmatrix} 1 & \mathbf{c} \\ \mathbf{c}^\dagger & z + (\mathbf{n} - \frac{1}{2}) \end{pmatrix}. \tag{157}$$

By employing the universal fusion formula,

$$\mathbf{L}_{\{1\}}^{[1]}(z - \tfrac{1}{2})\,\mathbf{L}_{\{2\}}^{[2]}(z - \lambda + \tfrac{1}{2}) = \mathbf{S}\,\mathbf{L}_\lambda^{[1]}(z)\,\mathbf{K}^{[2]}\,\mathbf{S}^{-1}, \tag{158}$$

we readily derive the $\mathfrak{gl}(1|1)$-invariant Lax operator which takes the form

$$\mathbf{L}_\lambda^{[1]}(z) = \begin{pmatrix} z + \lambda - (\mathbf{n}_1 - \frac{1}{2}) & -2\lambda\mathbf{c}_1^\dagger \\ -\mathbf{c}_1 & z - \lambda - (\mathbf{n}_1 - \frac{1}{2}) \end{pmatrix}, \tag{159}$$

where

$$\mathbf{K}^{[2]} = \begin{pmatrix} 1 & \mathbf{c}_2 \\ 0 & 1 \end{pmatrix}, \qquad \mathbf{S} = \exp\left(\mathbf{c}_1^\dagger \mathbf{c}_2^\dagger\right), \tag{160}$$

have been used. To match the canonical form of Eq. (40), given by

$$\mathbf{L}_{\{1,2\}}(z) = \begin{pmatrix} z - \mathbf{J}^{11} & \mathbf{J}^{21} \\ -\mathbf{J}^{12} & z + \mathbf{J}^{22} \end{pmatrix}, \tag{161}$$

the $\mathfrak{gl}(1|1)$ super spin generators $\mathbf{J}^{ab}$ are identified as

$$\mathbf{J}^{11} = j_1 + \mathbf{c}_1^\dagger \mathbf{c}_1, \qquad\qquad \mathbf{J}^{12} = (j_1 + j_2 - \lambda)\mathbf{c}_1^\dagger, \tag{162}$$

$$\mathbf{J}^{21} = \mathbf{c}_1, \qquad\qquad \mathbf{J}^{22} = j_2 - \lambda - \mathbf{c}_1^\dagger \mathbf{c}_1, \tag{163}$$

together with the following constraint on the $\mathfrak{gl}(1)$ scalars

$$j_1 = -\tfrac{1}{2} - \lambda, \qquad j_2 = \tfrac{1}{2}. \tag{164}$$

Therefore, $\mathbf{L}_\lambda(z)$ belongs to the two-dimensional representation $\mathscr{V}_\lambda$, with $\lambda$ being the central charge. A comparison with Eq. (156) shows that $\mathscr{V}_\lambda \equiv \langle 1, 2\lambda \rangle$. In terms of the fermionic algebra, operator $\mathbf{L}_\lambda(z)$ admits an expansion

$$\mathbf{L}_\lambda(z) = (-1)^b E^{ab} \circledast \mathbf{L}^{ab}(\lambda) = z + 2\lambda\, c\, \mathbf{c}^\dagger + c^\dagger\, \mathbf{c} - 2\lambda n - \mathbf{n}. \tag{165}$$

Recall that for $\lambda = 0$, module $\mathscr{V}_\lambda$ becomes an atypical indecomposable representation (a short multiplet) which is no longer irreducible; it contains a one-dimensional invariant subspace corresponding to the Fock vacuum. However, these exceptional instances do not seem to be relevant in the context of boundary-driven spin chains.

We moreover wish to emphasize that prescription (134) provides an explicit oscillator realization of $\mathfrak{gl}(n|m)$ Lie superalgebras. Let us consider the $\mathfrak{gl}(2|1)$ case as an example, and fixing the grading to $\otimes\!\!-\!\!\otimes$. The bosonic subalgebra is a direct sum $\mathfrak{gl}(2) \oplus \mathfrak{u}(1)$, and is spanned by the $\mathfrak{gl}(2)$ generators (writing $\mathbf{n}_i = \mathbf{c}_i^\dagger \mathbf{c}_i$)

$$\begin{aligned} \widehat{\mathbf{J}}^{11} &= \mathbf{J}_1^{11} + \mathbf{n}_1, & \widehat{\mathbf{J}}^{13} &= \mathbf{J}_1^{13} + \mathbf{c}_1^\dagger \mathbf{c}_2, \\ \widehat{\mathbf{J}}^{33} &= \mathbf{J}_1^{33} + \mathbf{n}_2, & \widehat{\mathbf{J}}^{31} &= \mathbf{J}_1^{31} + \mathbf{c}_2^\dagger \mathbf{c}_1, \end{aligned} \tag{166}$$

and the $\mathfrak{u}(1)$ generator

$$\widehat{\mathbf{J}}^{22} = j_2 - \mathbf{n}_1 - \mathbf{n}_2. \tag{167}$$

In addition, there are four fermionic charges which are parametrized as

$$\begin{aligned} \widehat{\mathbf{J}}^{12} &= \mathbf{c}_1^\dagger \widehat{\mathbf{J}}^{22} + \mathbf{J}_1^{11} \mathbf{c}_1^\dagger + \mathbf{J}_1^{13} \mathbf{c}_2^\dagger, & \widehat{\mathbf{J}}^{21} &= \mathbf{c}_1, \\ \widehat{\mathbf{J}}^{32} &= \mathbf{c}_2^\dagger \widehat{\mathbf{J}}^{22} + \mathbf{J}_1^{31} \mathbf{c}_1^\dagger + \mathbf{J}_1^{33} \mathbf{c}_2^\dagger, & \widehat{\mathbf{J}}^{23} &= \mathbf{c}_2. \end{aligned} \tag{168}$$

The $\mathfrak{sl}(1|2)$ case can be obtained by restricting $\mathbf{J}_1^{ab}$ to the $\mathfrak{sl}(2)$ spins acting on $\mathscr{V}_{j_1}$, while $j_2$ is the remaining Dynkin label.

# C Non-interacting fermions

In this section we provide the solution to the problem of boundary-driven non-interacting spinless fermions hopping on a one-dimensional lattice. The Hamiltonian of the model can be seen as a Yang–Baxter integrable spin chain invariant under $\mathfrak{gl}(1|1)$ Lie superalgebra.[25]

---

[25]The model can also be mapped to the XX Heisenberg spin chain Hamiltonian. In the spin picture, we deal with a model invariant under the $q$-deformed quantum symmetry $\mathscr{U}_q(\mathfrak{sl}(2))$ for the value of the deformation parameter $q = \mathrm{i}$.

In Section 5.2 we constructed the steady-state solutions for the simplest fermionic boundary reservoirs with equal coupling strengths. Our aim here is to demonstrate that the problem of free fermions represents a special case which even allows for solutions going beyond those discussed in Section 5.2. Quite remarkably, the *operator* Schmidt rank[26] of $\rho_\infty$ now equals 4 and thus does not depend on the system size. This is in stark contrast to the solutions pertaining to higher-rank symmetries which all exhibit Schmidt ranks which grow algebraically with system size. A finite Schmidt rank gives a strong indication that the problem of finding the steady states may be tractable by 'brute force', that is first by explicitly computing the null space of the Liouvillian generator $\mathscr{L}$ and subsequently analytically parametrizing the solution (e.g. with help of symbolic algebra routines). An obvious advantage of this approach is that it does not require any prior knowledge of the underlying algebraic structures. This allows allows to conveniently parametrize the solutions directly in terms of physical couplings attributed to the boundary reservoirs.

We shall provide an extended four-parametric set of solutions for the asymmetric driving

$$A_1 = \sqrt{g\,\zeta}\,\sigma_1^+, \qquad A_N = \sqrt{g/\zeta}\,\sigma^-, \tag{169}$$

involving coupling rate parameters $g, \zeta \in \mathbb{R}$, supplemented with two additional boundary external fields,

$$H_{\text{field}} = \frac{h_{\text{L}}}{2}\sigma_1^z + \frac{h_{\text{R}}}{2}\sigma_N^z, \tag{170}$$

of unequal magnitudes $h_{\text{L}}, h_{\text{R}} \in \mathbb{R}$. We notice that the solutions given below appear to lie outside of $\mathfrak{gl}(1|1)$-invariant Lax operators of Eqs. (68) and (69).[27]

The **L**-operator for the $\Omega$-amplitude is now formally linked to a two-dimensional auxiliary representation denoted by $\mathscr{V}_{\mathbf{u}}$. Here **u** is a four-component vector label which involves the boundary parameters, $\mathbf{u} = (g, \zeta, h_{\text{L}}, h_{\text{R}})$. In terms of Pauli matrices, the **L**-operator admits the expansion

$$\mathbf{L}_{\mathscr{V}_{\mathbf{u}}} = \frac{\sigma^0}{2} \otimes \begin{pmatrix} \zeta+1 & 0 \\ 0 & (\zeta-1)+(\tilde{h}_{\text{L}}-\zeta\,\tilde{h}_{\text{R}}) \end{pmatrix} + \frac{\sigma^z}{2} \otimes \begin{pmatrix} \zeta-1 & 0 \\ 0 & ig(1-\zeta)-(\tilde{h}_{\text{L}}+\zeta\,\tilde{h}_{\text{R}}) \end{pmatrix}$$
$$+ \sigma^- \otimes \begin{pmatrix} 0 & 0 \\ ig(\zeta^2+1)+\zeta\,\delta h & 0 \end{pmatrix} + \sigma^+ \otimes \begin{pmatrix} 0 & 1 \\ 0 & 0 \end{pmatrix}, \tag{171}$$

using shorthand notations $\tilde{h}_i = h_i - 1$ and $\delta h = h_{\text{L}} - h_{\text{R}}$. It can easily be verified that the **L**-operator provides a *multi-colored* family of commuting of transfer matrices,

$$T(\mathbf{u}) = \text{Tr}_{\mathscr{V}_{\mathbf{u}}} \mathbf{L}_{\mathscr{V}_{\mathbf{u}}} \otimes \cdots \otimes \mathbf{L}_{\mathscr{V}_{\mathbf{u}}}, \qquad [T(\mathbf{u}_1), T(\mathbf{u}_2)] = 0 \qquad \forall \mathbf{u}_1, \mathbf{u}_2 \in \mathbb{C}^4. \tag{172}$$

The involution property of $T(\mathbf{u})$ is ensured by the multi-colored 6-vertex $R$-matrix $\mathbf{R}_{\mathscr{V}_{\mathbf{u}_1}\mathscr{V}_{\mathbf{u}_2}}$ which operates on $\mathscr{V}_{\mathbf{u}_1} \otimes \mathscr{V}_{\mathbf{u}_2}$,

$$\mathbf{R}_{\mathscr{V}_{\mathbf{u}_1}\mathscr{V}_{\mathbf{u}_2}} = \begin{pmatrix} a_1(\mathbf{u}_1,\mathbf{u}_2) & 0 & 0 & 0 \\ 0 & b_1(\mathbf{u}_1,\mathbf{u}_2) & c_1(\mathbf{u}_1,\mathbf{u}_2) & 0 \\ 0 & c_2(\mathbf{u}_1,\mathbf{u}_2) & b_2(\mathbf{u}_1,\mathbf{u}_2) & 0 \\ 0 & 0 & 0 & a_2(\mathbf{u}_1,\mathbf{u}_2) \end{pmatrix}, \tag{173}$$

intertwining two copies of auxiliary spaces $\mathscr{V}_{\mathbf{u}}$ associated to a pair of **L**-operators $\mathbf{L}_{\mathscr{V}_{\mathbf{u}_i}}$ acting on $\mathbb{C}^2 \otimes \mathscr{V}_{\mathbf{u}_i}$,

$$\mathbf{R}_{\mathscr{V}_{\mathbf{u}_1}\mathscr{V}_{\mathbf{u}_2}} \mathbf{L}_{\mathscr{V}_{\mathbf{u}_1}} \mathbf{L}_{\mathscr{V}_{\mathbf{u}_2}} = \mathbf{L}_{\mathscr{V}_{\mathbf{u}_2}} \mathbf{L}_{\mathscr{V}_{\mathbf{u}_1}} \mathbf{R}_{\mathscr{V}_{\mathbf{u}_1}\mathscr{V}_{\mathbf{u}_2}}. \tag{174}$$

---

[26]The operator analogue of the Schmidt rank characterizes a degree of bipartite entanglement of a mixed quantum state. In the language of matrix-product states it coincides with the bond dimension.

[27]In practice it turns out that free fermions allow even more general types of non-perturbative integrable boundaries than those considered here.

Moreover, it can easily be verified that the **R**-matrix embedded into a three-fold tensor space obeys the Yang–Baxter equation

$$\mathbf{R}_{\mathscr{V}_{\mathbf{u}_1}\mathscr{V}_{\mathbf{u}_2}}\mathbf{R}_{\mathscr{V}_{\mathbf{u}_1}\mathscr{V}_{\mathbf{u}_3}}\mathbf{R}_{\mathscr{V}_{\mathbf{u}_2}\mathscr{V}_{\mathbf{u}_3}} = \mathbf{R}_{\mathscr{V}_{\mathbf{u}_2}\mathscr{V}_{\mathbf{u}_3}}\mathbf{R}_{\mathscr{V}_{\mathbf{u}_1}\mathscr{V}_{\mathbf{u}_3}}\mathbf{R}_{\mathscr{V}_{\mathbf{u}_1}\mathscr{V}_{\mathbf{u}_2}}. \tag{175}$$

The amplitudes (Boltzmann weights) of the *R*-matrix read explicitly

$$
\begin{aligned}
a_1(\mathbf{u}_1,\mathbf{u}_2) &= \zeta_1\zeta_2\, g_1 + g_2 - \mathrm{i}\,\zeta_2(h_{\mathrm{L},1} - h_{\mathrm{R},2}),\\
a_2(\mathbf{u}_1,\mathbf{u}_2) &= g_1 + \zeta_1\zeta_2 g_2 - \mathrm{i}\,\zeta_1(h_{\mathrm{L},2} - h_{\mathrm{R},1}),\\
b_1(\mathbf{u}_1,\mathbf{u}_2) &= \zeta_1 g_2 - \zeta_2 g_1 - \mathrm{i}\,\zeta_1\zeta_2(h_{\mathrm{R},1} - h_{\mathrm{R},2}),\\
b_2(\mathbf{u}_1,\mathbf{u}_2) &= \zeta_1 g_1 - \zeta_2 g_2 - \mathrm{i}(h_{\mathrm{L},1} - h_{\mathrm{L},2}),\\
c_1(\mathbf{u}_1,\mathbf{u}_2) &= (1 + \zeta_2^2)g_2 - \mathrm{i}\,\alpha_2(h_{\mathrm{L},2} - h_{\mathrm{R},2}),\\
c_2(\mathbf{u}_1,\mathbf{u}_2) &= (1 + \zeta_1^2)g_1 - \mathrm{i}\,\alpha_1(h_{\mathrm{L},1} - h_{\mathrm{R},1}),
\end{aligned}
\tag{176}
$$

and satisfy the *free fermion condition* [154],

$$a_1 a_2 + b_1 b_2 = c_1 c_2. \tag{177}$$

The differential Yang–Baxter relation on $\mathbb{C}^2 \otimes \mathbb{C}^2$ yields the standard form of the Sutherland equation,

$$\left[h^{1|1}, \mathbf{L}_{\mathscr{V}_{\mathbf{u}}} \otimes \mathbf{L}_{\mathscr{V}_{\mathbf{u}}}\right] = \mathbf{L}_{\mathscr{V}_{\mathbf{u}}} \otimes \widetilde{\mathbf{L}}_{\mathscr{V}_{\mathbf{u}}} - \widetilde{\mathbf{L}}_{\mathscr{V}_{\mathbf{u}}} \otimes \mathbf{L}_{\mathscr{V}_{\mathbf{u}}}, \tag{178}$$

where

$$\widetilde{\mathbf{L}}_{\mathscr{V}_{\mathbf{u}}} = \tfrac{1}{2}(\mathrm{i}\,g - \zeta h_{\mathrm{R}})\sigma^0 \otimes \boldsymbol{\sigma}^0 - \tfrac{1}{2}(\mathrm{i}\,g\zeta + h_{\mathrm{L}})\sigma^z \otimes \boldsymbol{\sigma}^z. \tag{179}$$

By expanding it to the entire spin chain, we obtain the action of the unitary part $\mathscr{L}_0$ which in distinction to the canonical solutions discussed in the paper this time (assuming non-vanishing $\delta h$) acquires an additional term,

$$[H, \Omega_N(\mathbf{u})] = \tilde{g}\begin{pmatrix} \zeta\,\delta h + 1 & 0 \\ 0 & \zeta \end{pmatrix} \otimes \Omega_{N-1}(\mathbf{u}) + \tilde{g}\,\Omega_{N-1}(\mathbf{u}) \otimes \begin{pmatrix} 1 & 0 \\ 0 & -\zeta - \delta h \end{pmatrix} + \delta h\,\Omega_N(\mathbf{u}). \tag{180}$$

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
