# Peer review of "Dissipation-driven integrable fermionic systems: from graded Yangians to exact nonequilibrium steady states"

_SciPost Physics, doi:SciPost Phys. 3, 031 (2017)_

## Round 2 · Referee Report · Anonymous (Referee 1) · 2017-3-23

Strengths

  1. carefully written
  2. technically strong

Weaknesses

  1. no clear separation between innovations brought by this paper, and results previously obtained by others
  2. the bibliography is not selective at all, and does not orient the reader towards references that are relevant to appreciate the results of this paper
  3. no new result about any concrete physical observable
  4. no pictures (while algebraic manipulations of Matrix Product Operators/monodromy matrices are sometimes best illustrated with diagrams)

Report

The author investigates algebraic structures that underly the study of steady states in a class of spin chains evolving with Lindblad dynamics. The dynamics is generated by an integrable hamiltonian in the bulk, and dissipators that act only on the boundaries of the chain. This class of exactly solvable models has been studied extensively in the past years (Refs. [72-80]) with techniques that exploit both integrability of the Hamiltonian and the possibility of finding exact steady states in the form of matrix product states (dating back to Refs. [47,48]). This paper provides a generalization of those works in the case of $\mathfrak{gl}(n|m)$ super-spin chains, and discusses connections between various algebraic structures that appear in this model.

In my opinion, even though it is carefully written, the manuscript is difficult to read. It is addressed only to an expert audience, with a sharp knowledge of the existing literature. The main claim of the author is that his framework is more general than the existing one, and that it offers a unified perspective on all previous results, such as the ones of Refs. [72-80]. But the paper is written in such a way that it is very difficult to distinguish the new ideas from those that are simply taken from Refs. [72-80]. As a consequence, the relevance of the whole construction has remained quite unclear to me. In its present form, I cannot recommend this paper for publication in Scipost.

Requested changes

I have several objections to the way the paper is presented.

  1. The main problem is, in my opinion, that the formalism of integrable spin chains driven by boundary dissipators is not properly reviewed in the introduction, or in a separate section at the beginning of the paper. It should be explained clearly that this problem was already solved, for instance, for the XXZ chain, and that such key objects as the matrix product operator $\Omega_N$ were introduced in that context. This pre-existing formalism should be recalled, and the reader should be oriented towards relevant (selected) references.

Instead, the author mixes the introduction of the model he’s studying (a spin chain with $\mathbb{C}^{n|m}$ degrees of freedom) with important pre-existing ideas and formalism, without properly referencing those. This generates confusion, and for a non-expert reader, it is hard to see what the new material is here.

I think it would be much better to have two clearly separated sections, one with a review of previous results in standard spin chains (i.e. not $\mathbb{Z}_2$ graded), and the other with a clear presentation of the new ones in the context of $\mathbb{Z}_2$ grading.

  1. A related problem is the fact that, apart from the last paragraph and the outline, the introduction is not very informative, and does not introduce the true topic of the paper to the reader. Instead, it contains a large number of citations to works that are only very remotely connected to the content of the manuscript (the very first paragraph, for instance, contains no less 36 references to papers that have essentially nothing to do with the present one).

I would find it helpful if the introduction and the list of references were more focused on the true topic of the paper, as in the last paragraph ‘’In this work we shall focus exclusively on formal algebraic aspects related to a class of boundary-driven integrable spin chains. […]’’. The introduction could focus on the presentation of this class of models, and on some of the key ideas that have appeared in that context, like the factorization property $\rho_\infty = \Omega_N \Omega^\dagger_N$, that are really needed later in the paper.

  1. There is some confusion in the way the discussion is organized between sections 2 and 3. Some of the notations for $\mathbb{Z}_2$ grading are being used in section 2 (for instance in Eq. (6)) without a definition; they are defined only later in section 3 (or also in appendix A).

  2. When reading all the algebraic relations in the paper, the reader always needs to be careful about whether operators act in physical space or in auxiliary space. This is of course inherent to the topic itself, but usually, drawing diagrams helps a lot. I think it would really help the reader follow the logic of the paper if the author could represent the key equations diagrammatically.

  • validity: high
  • significance: ok
  • originality: ok
  • clarity: low
  • formatting: good
  • grammar: excellent

Author:  Enej Ilievski  on 2017-05-09  [id 132]

(in reply to Report 1 on 2017-03-23)

We thank the referee for reading the manuscript and for suggesting how to improve it. Let us in turn address the most important points:

(I) Let us begin by remarking that there is no bona fide formalism or framework for solving this particular class of problems available at the moment. This has been for us the main motivation for continuing the research in this direction. It also deserves to be to pointed out that references [72-80] differ quite drastically in their approach and ideology from our work. In particular, these paper do not even make use of quantum symmetries and other integrability concepts. Quantum groups or Lax formulation have been first used in our previous work [81,82], when addressing the particular cases of the anisotropic Heisenberg model and su(3) spin chain, which have unfortunately been overlooked by the referee.

(II) The meaning of "It is very difficult to distinguish the new ideas from those that are simply taken from Refs. [72-80]" is obscure to us. In this work, we operate with formal concepts and algebraic structures which have not been used or discussed before in the scope of this application, modulo to some extent the above-mentioned refs. [81.82]. The solutions derived in our manuscript are nonetheless new, and arguably not just a straightforward generalization of what has been known before.

(III) We disagree with the referee that having two separate sections, one for the non-graded and another for the graded algebras, would be better. The former are just a special case of the latter when the grading is trivial. We find it much better to treat everything on equal footing. Apart from this, we wish to inform the referee that the su(n) spin chains with this class of integrable dissipative boundaries have been solved before.

(IV) We have employed bold-faced symbols to explicitly distinguish operators which act non-trivially in auxiliary spaces from those acting only in the physical space. This convention has been used already in many previous works, and we do not see nay ambiguity with it. Regardless of this, we often explicitly emphasize in the text on which spaces various Lax operators operate on.

(V) Unfortunately, we cannot provide any further insight or explanations regarding the factorization property. Nevertheless, saying that "the factorization property is needed in the paper" is slightly confusing. The property is apparently a feature (or a consequence) of the chosen integrable dissipative boundaries, but an apriori justification remains unknown. The property can be for instance inferred by inspecting the structure of the solutions by explicitly diagonalizing small instances on a computer.

List of changes: - Improved second part of the introduction to emphasize the new contributions. - The introduction of graded vectors spaces now appears in section 2.2, before introducing the graded Yang-Baxter equation and other integrability-related concepts. - We have followed the referee's advice and included a few diagrams and graphical representations of key algebraic identities, hoping that the readers will find them helpful.

---

## Round 2 · Referee Report · Anonymous (Referee 2) · 2017-4-3

Strengths

the paper is technically precise
the results are correct
the paper provides an overview of the subject
the previous results are put into a nice algebraic structure

Weaknesses

integrability is not reviewed enough, making the paper hard to read; section 2 should be improved
it is not clear what is new

Report

This is about the recent high-profile works on building exact nonequilibrium steady states in driven quantum integrable spin chains. The manuscript reviews some of the work, and puts it in a common algebraic framework. This is a very interesting endeavour, it is important to clarify the underlying algebraic structure in order to eventually connects with advanced concepts of integrability. The structure identified is based on Yangians.

Since this is mostly reviewing previous works, the technical accuracy of the results is not in doubt. The most important thing is to put the results within a framework. I think this is relatively well done in sections 3 and 4. But section 2 needs improvement, I would say especially section 2.4. I think a lot of knowledge of the structures of integrability is assumed, and it makes the paper rather hard to read.

As specific examples, what is $\mathcal{O}^\nu$ on page 9? A linear space of an operator? The term "$\Omega$-amplitude" is used without really being defined. Also, the signature $n|m$ is explained in section 3 but used already in 2.4. Also, footnote (4) is unclear (I don't know what it means). These are small points, but in addition to these, overall I would be happy to see the discussion improved and clarified in section 2.

Another point is that it is not entirely clear what is new, and what is not new. Emphasizing the new structures identified would be good.

Requested changes

See report.

  • validity: high
  • significance: ok
  • originality: ok
  • clarity: low
  • formatting: good
  • grammar: excellent

Author:  Enej Ilievski  on 2017-05-09  [id 131]

(in reply to Report 2 on 2017-04-03)

We thank the referee for evaluating our manuscript and for their suggestions. In regard to their criticism, we wish to make the following remarks.

(I) The claim that the manuscript is mostly reviewing previous results is inappropriate. We would like to stress that the class of steady-state solution for the driven integrable chains which pertain to higher-rank symmetries has not been obtained prior to our work. Moreover, here we present for the first time the canonical construction based on the representation theory of Yangians, uncover the connection to Baxter's operators, and finally extend/adapt the construction to fermionic models which require non-unitary gl(1|1) modules. Besides all of these, we also presented an extended class of solutions for then boundary-driven free fermions - which relate to multi-colored Yang-Baxter algebras - also not known prior to our work.

(II) While we agree with the referee that technical accuracy is not in question, the reason surely is not because of "reviewing previous works". Instead, it is because we have confirmed our calculations with symbolic algebra computations and also explicitly constructed the steady-states on a computer.

(III) We had trouble understanding what the referee meant by saying "integrability is not reviewed enough". In the absence of more concrete suggestions, it is difficult to make improvements in this department. Let us nonetheless point out that Yang-Baxter equation (in its graded version), along with its differential form, are the only algebraic relations required to reproduce the results. The remaining part is to make use of universal factorization formulas and allow for non-unitary representations of the underlying algebras. These aspects are introduced in the main text, with further details postponed to appendices. Knowing the logic behind various other Bethe Ansatz techniques is not significant for the results in this paper.

(IV) The amplitude operator is implicitly defined in equation (8), and explicitly shortly after in equation (10).

(V) The main contribution of our work is summarized already in the abstract. Additional information can be found in the introduction, which has now been improved.

List of changes: - Section 3.4 on integrable boundary conditions has been revised and improved. - A new Section 2.2 has been added (first part of Section 3 in the previous version), where some essential definitions and notations regarding graded vector spaces can be found. - The second part of the introduction has been rewritten in order to better emphasize the new contributions of this work.

---

## Round 2 · Referee Report · Anonymous (Referee 3) · 2017-4-11

Strengths

The subject and the aim of the paper are interesting.

Weaknesses

The presentation of several main objects is poor and somewhere uncertain.

Report

The author address the interesting issue to provide an algebraic framework to the derivation of the steady-state density matrix for some classes of open quantum systems with dissipation driven localized on the boundaries.

The approach presented rely on the integrable structure of the Hamiltonian associated to the open quantum system. More in detail, the author considers this problem for a class of models associated to graded Sl(n|m) Hamiltonians. The steady-state density matrix is then given in a factorized form as a product of two conjugate "amplitude operators" which in turns are written as the expectation values of a product of "Partonic" Lax operators, of references [85-88], over a so-called "vacuum state".

The manuscript is quite well written in its introduction, conclusion and section 3 and also in the part where are reproduced the graded representations and the "Partonic" Lax operators. On the other hand, I find that the presentation of several main objects is poor and somewhere uncertain leaving the reader missing of several operative definitions and so unable to reproduce by himself the material presented in the manuscript.

Let me start from my main problem, that is the definition of the vacuum states (one of the two fundamental ingredients to characterize the steady-state density matrix in the author's proposal).

The most clear definition that I have found of it is the following general statement given at page 5 of section 2:

"In all the instances addressed in this work, the vacuum state pertains to a product of highest- or lowest-weight vectors from irreducible spaces associated to the auxiliary algebra of the L-operator."

The problem for me is that in section 4 where concrete examples of open quantum models are taken the vacuum state is not explicitly written; the author should present just one equation where it is defined for any considered model. For example one should have not only the definition of the Lax operator (45) but also of the vacuum state to show that the condition (47) allows to satisfy the equation (19); and the same is required for all the other cases presented in the manuscript.

There are moreover unclear notations in section 2, just to make some examples, the definition of the Hamiltonian above equation (6) is not explicit, in equation (7) it is written a graded tensor product of L matrix but the number of them is not explicitly written; are they N like the number of quantum site of the chain? The same problem is in equation (17). In equation (24) it is unclear the index k which values takes.

Once these clarifications are done there are still some conceptual points that have to be clarified. The Hamiltonian of the models considered in this manuscripts should be open ones, no closing integrable conditions are given but a priori it should be very important to clarify if some integrable boundary conditions are considered. Indeed, the conserved charges of the model depends from these boundary conditions and they should play an important role in the determination of the steady-state density matrix according to the author statement at page 5 of section 2:
" This suggests that the algebra of (possibly non-local) commuting operators belonging to H_{n|m} might be a good place to look for an appropriate Ω-amplitude of Eq. (5). "

and to the discussion in section 2.4.

Furthermore, in section 5 the author seems to justify his introduction of the "vacuum Q-operator" (expectation value on the "vacuum state" of a product of Lax matrices) as for open quantum system "taking (super) traces over auxiliary spaces cease to make sense " for the absence of translation invariance. Well here it is important to remark that the integrable structure of an open quantum system codified by a reflection equation and its dual allow to overcome such a problem. This is already clear in the seminal work of Sklyanin "Boundary conditions for integrable quantum systems " of the 1988.

I want also to mention that at least a couple of statements in the paper are out of place, at page 3 the author write:

"However, to best of our knowledge, these objects have mostly been used as a formal tool in the Bethe Ansatz diagonalization techniques, while their role in concrete physics applications remains rather unclear"

there are several physical applications of the knowledge of the Baxter Q-operator, for example it is an intermediate step in the computation of correlation functions and dynamical structure factors.

at page 19 of the section 5, referring to the Baxter Q-operator the author write:

"The concept was resurrected more than two decades later in the context of CFTs"

well between the seminal work of Baxter of the 1972 and the series of works of Bazhanov, Lukyanov and Zamolodchikov there is a vast literature in integrable quantum models about the Baxter Q-operator, just to make an example let me mention the connection that it allows to establish between two a priori very different models like the 6-vertex model and the chiral Potts model, as pioneered in the paper of Bazhanov and Stroganov of the 1989.

Requested changes

In order the paper can be accepted for publication on SciPost I suggest that the author implement the clarifications above mentioned and make more explicit some of the content of section 2.

  • validity: -
  • significance: good
  • originality: ok
  • clarity: low
  • formatting: good
  • grammar: good

Author:  Enej Ilievski  on 2017-05-09  [id 130]

(in reply to Report 3 on 2017-04-11)

We thank the referee for the critical reading of our manuscript, and many helpful remarks.

(I) The auxiliary vacua are always factorized in terms of extremal-weight states belonging to the irreducible subspaces of the auxiliary algebra. This can be considered as a definition, and is sufficient to unambiguously define the solutions. Since Lax operators carry representation labels which provide information about the types of auxiliary spaces, there is no need to write them out explicitly.

(II) Our justification for employing vacuum Q-operators does not come from the absence of translational symmetry per se. We are familiar with Sklyanin's construction, but we doubt that it can be applied in the present situation. Unlike in our construction, commuting transfer matrices which are derived from reflection algebras still employ auxiliary traces. On the other hand, the boundary reflections represent unitary processes, in start contrast to Lindblad jump operators which represent non-unitary and non-Hamiltonian processes.

(III) In our opinion there can be no confusion about the boundary conditions. The nonequilibrium protocol with boundary driving scrutinized in the manuscript only makes sense for Hamiltonians with open boundaries. One should also note that the local structure of the conservation laws is insensitive to the type of (Hamiltonian) boundary conditions. The global conservation laws on the other hand exactly commute with the Hamiltonian and hence are not what we were after (noticing that the fixed-point density matrices should not be individually preserved by the Hamiltonian and dissipative part of Lindbladian dynamics).

List of changes: - The meaning of auxiliary vacuum states has been more explicitly clarified. We have emphasized that the vacuum state for a given solution is uniquely defined by specifying the types of auxiliary spaces for the two-row Lax operator. - We have emphasized that our models are considered with open boundary conditions in two distinct ways: as quantum lattice model without assuming periodic boundary conditions, and as a quantum system which experience dissipation and thus evolve non-unitarily. - Added reference to Bazhanov and Stroganov (1989).

---

## Round 2 · Referee Report · Anonymous (Referee 2) · 2017-6-8

Strengths

1- the paper is technically precise 2- the results are correct 3- the paper provides an overview of the subject 4- the previous results are put into a nice algebraic structure

Weaknesses

  • The paper is hard to read.

Report

This is about the recent high-profile works on building exact nonequilibrium steady states in driven quantum integrable spin chains. The manuscript reviews some of the work, and puts it in a common algebraic framework. The structure identified is based on Yangians.

Again, this is an interesting work, it is important to clarify the underlying algebraic structure in order to eventually connects with advanced concepts of integrability. There is a lot of material, and some interesting ideas.

However, there is no way out of the fact that the paper is extremely hard to read. Many aspects are developed in different parts, and it is easy to get lost (in the notation, in the concepts, in what it is that the author is trying to do). Perhaps, if there was an overview section where all aspects were put together, all explained, so that we could have a big picture, that would be better. But I don't know if this is possible at all. I understand that the author has made an effort to improve the text, it is more readable then previous version (I think everything is defined, at least somewhere, even if sometimes we have to search for the definition).

Here are some little things that I found unclear / inaccurate:

p 7, bottom: can the author be more precise about what a "representation label" is.

p10: the dissipator acting on Omega cancels right the hand side of (20) (that is, it gives a term that equates the negative of rhs of (20); it does not annihilate rhs of (20) in the sense that acting on rhs of (20) with it does not give zero)

p 9 before eq 18: the hamiltonian density does not coincide with the permutation, as in 18 there is an extra factor of derivative of R.

p9: "the differential equation (19)" : should be (18) instead? "A general solution to eq 18" should be eq 17 instead?

p9 in paragraph after eq 18: which one is physical space and which is auxiliary space? please specify

p9 just after eq 19: in expression of ${\cal H}_{\rm aux}$ why is index now $n$ instead of $n+m$?

p10: what does it mean that two copies of a Hilbert space are "mutually conjugate" (just above eq 21)? A Hilbert space is either isomorphic or not to another Hilbert space, but I don't see how it can be conjugate.

p 11 top: "pertains to the conjugate dual" -> does it really mean "is the conjugate dual"? I don't understand the use of "pertain".

p 11 bottom: "a general solution of condition (19)"... but (19) is not a condition, it is a definition.

p12 fig 5: it is a bit strange to put 2A where it is: why not a factor 2 for this term instead?

p14: why are level-0 grassmann numbers? they seem to commute in eq 32.

p14: how is eq 33 linear in z? there is z^{-1}, so it is rather linear in z^{-1}

p 20, footnote 13: where is the Jordan-Wigner transformation used? It is not clear in the main text where it would be involved. Why is it involved in the dissipator?

Requested changes

At this moment I do not know what changes to ask, even though the reading is difficult. I'd say, then, only clarify the small things listed in the report and repeated here:

p 7, bottom: can the author be more precise about what a "representation label" is.

p10: the dissipator acting on Omega cancels right the hand side of (20) (that is, it gives a term that equates the negative of rhs of (20); it does not annihilate rhs of (20) in the sense that acting on rhs of (20) with it does not give zero)

p 9 before eq 18: the hamiltonian density does not coincide with the permutation, as in 18 there is an extra factor of derivative of R.

p9: "the differential equation (19)" : should be (18) instead? "A general solution to eq 18" should be eq 17 instead?

p9 in paragraph after eq 18: which one is physical space and which is auxiliary space? please specify

p9 just after eq 19: in expression of ${\cal H}_{\rm aux}$ why is index now $n$ instead of $n+m$?

p10: what does it mean that two copies of a Hilbert space are "mutually conjugate" (just above eq 21)? A Hilbert space is either isomorphic or not to another Hilbert space, but I don't see how it can be conjugate.

p 11 top: "pertains to the conjugate dual" -> does it really mean "is the conjugate dual"? I don't understand the use of "pertain".

p 11 bottom: "a general solution of condition (19)"... but (19) is not a condition, it is a definition.

p12 fig 5: it is a bit strange to put 2A where it is: why not a factor 2 for this term instead?

p14: why are level-0 grassmann numbers? they seem to commute in eq 32.

p14: how is eq 33 linear in z? there is z^{-1}, so it is rather linear in z^{-1}

p 20, footnote 13: where is the Jordan-Wigner transformation used? It is not clear in the main text where it would be involved. Why is it involved in the dissipator?

---

## Round 3 · Referee Report · Anonymous (Referee 5) · 2017-6-6

Strengths

1. technically strong

Weaknesses

1. still no clear separation between new results and the rest

Report

The author has made modifications to his manuscript. He has reorganized his sections 1, 2 and 3,
and added diagrams to illustrate the main equations. While I find that the quality of the presentation
has improved, the main issue of the manuscript remains: it is still not clear to me what exactly is the novelty
brought by the paper.

What are the important results? What are the equations that the author wants to emphasize? This should
be made obvious to the reader. It should be explained clearly, both in the introduction and in the conclusion,
or perhaps in a separate section that could summarize the results.

For a start, the paper could easily gain clarity if the introduction was focused on the content of the paper,
instead of consisting in a long list of remotely connected topics (e.g., is it really necessary to start with the
traditional sentence about «remarkable progress in experiments in cold atoms», while the
paper is actually about graded Yangians? Wouldn’t it be more helpful and less confusing for the reader
to get to the point more quickly?).

Overall, the new results must be exposed clearly, and contrasted with the ones that are already available in the literature.
For instance, in his reply to my report, the author writes

« We disagree with the referee that having two separate sections, one for the non-graded and another for the graded algebras, would be better. The former are just a special case of the latter when the grading is trivial. We find it much better to treat everything on equal footing. Apart from this, we wish to inform the referee that the su(n) spin chains with this class of integrable dissipative boundaries have been solved before. »

But this is precisely what I am talking about! The fact that the author presents the graded and non-graded cases together,
as if the whole thing was his own contribution, is confusing because the non-expert reader cannot possibly see what is new and
what isn’t. A better, clearer, more reader-friendly presentation, would be to start by recalling the existing result for the non-graded
case, and then explain why it is possible and interesting to generalize this pre-existing result to the graded case.

Since the author did not yet make all the necessary changes to make his manuscript clearer and more accessible,
I am still unsure about the relevance of the results it contains. I am still unable to recommend its publication
on Scipost.

Requested changes

1. re-organize the paper, in particular with a separate section containing a short review of the existing results. Then, in later sections, contrast those results with the new ones in the paper
2. the new version contains many typos, missing words, etc. Please proofread and make corrections
3. I suggest that the introduction be rewritten. Make it less broad, get to the point more quickly.

---

## Round 3 · Referee Report · Anonymous (Referee 3) · 2017-6-30

Strengths

the paper is technically precise
the results are correct
the paper provides an overview of the subject
the previous results are put into a nice algebraic structure

Weaknesses

The paper is hard to read.

Report

This is about the recent high-profile works on building exact nonequilibrium steady states in driven quantum integrable spin chains. The manuscript reviews some of the work, and puts it in a common algebraic framework. The structure identified is based on Yangians.

Again, this is an interesting work, it is important to clarify the underlying algebraic structure in order to eventually connects with advanced concepts of integrability. There is a lot of material, and a lot of interesting ideas.

However, there is no way out of the fact that the paper is somewhat hard to read. Many aspects are developed in different parts, and it is easy to get lost (in the notation, in the concepts, in what it is that the author is trying to do). Perhaps, it it were possible, if there was a section where all aspects were put together in a simple way, the fundamental ideas being explained, so that we could have a big picture, that would be helpful. But I understand that the author has made an effort to improve the text, and this is certainly improved on the first version (I think everything is defined, at least somewhere, even if sometimes we have to search a little bit for the definition).

Here are some small things that I found unclear / inaccurate:

p 7, bottom: can the author be more precise about what a "representation label" is.

p10: the dissipator acting on Omega cancels right the hand side of (20) (that is, it gives a term that equates the negative of rhs of (20); it does not annihilate rhs of (20) in the sense that acting on rhs of (20) with it does not give zero)

p 9 before eq 18: the hamiltonian density does not coincide with the permutation, as in 18 there is an extra factor of derivative of R.

p9: "the differential equation (19)" : should be (18) instead? "A general solution to eq 18" should be eq 17 instead?

p9 in paragraph after eq 18: which one is physical space and which is auxiliary space? please specify

p9 just after eq 19: in expression of ${\cal H}_{\rm aux}$ why is index now $n$ instead of $n+m$?

p10: what does it mean that two copies of a Hilbert space are "mutually conjugate" (just above eq 21)? A Hilbert space is either isomorphic or not to another Hilbert space, but I don't see how it can be conjugate.

p 11 top: "pertains to the conjugate dual" -> does it really mean "is the conjugate dual"? I don't understand the use of "pertain".

p 11 bottom: "a general solution of condition (19)"... but (19) is not a condition, it is a definition.

p12 fig 5: it is a bit strange to put 2A where it is: why not a factor 2 for this term instead?

p14: why are level-0 grassmann numbers? they seem to commute in eq 32.

p14: how is eq 33 linear in z? there is z^{-1}, so it is rather linear in z^{-1}

p 20, footnote 13: where is the Jordan-Wigner transformation used? It is not clear in the main text where it would be involved. Why is it involved in the dissipator?

Requested changes

If possible, a section where all aspects are put together in a simple way, the fundamental ideas and basic formulae being explained, so that we could have a big picture.

---

## Round 3 · Referee Report · Anonymous (Referee 2) · 2017-7-3

Strengths

.

Weaknesses

.

Report

Dear Editor,

I have read the replay of the author, the modifications made and read once again the manuscript. While I can find some improvements my previous judgment of the main manuscript problem stays:

"I find that the presentation of several main objects is poor and somewhere uncertain leaving the reader missing of several operative definitions and so unable to reproduce by himself the material presented in the manuscript."

Once again for me the author has missed the opportunity to use the examples of open quantum models presented (now) in Section 5 of its manuscript to clarify its definitions.

In point I) of his replay for the definition of the "vacua state" the author use some general statement speaking of "extremal-weight states belonging to the irreducible subspaces" and arguing that "This can be considered as a definition, and is sufficient to unambiguously define the solutions.".
Well these "extremal-weight states" are not clear for me and the author should identify them explicitly for any examples of open quantum models presented in Section 5.

Requested changes

I think that this is a minimal clarification required to make the material presented in this manuscript understandable for a general audiance and so in the interest of SciPost and of the author himself.

---

## Round 3 · Author Response

We are thankful to all three referees for their reports and many useful remarks. We have prepared a revised version of the manuscript by taking into account various suggestions from all the referees.

---

## Round 4 · Referee Report · Anonymous · 2017-9-13

Report

Dear Editor,

as required in my previous rapport, the author has made some modifications going in the direction of clarification in particular in the section 5.

I have not others important objections for the publication of this manuscript.

Best regards,

---

## Round 4 · Author Response

We again thank all the referees for carefully reading the manuscript and for providing valuable feedback. We have taken their suggestions into account and accordingly prepared a revised version of the manuscript. The main changes are summarized along with our response to the criticism.

1- The primary purpose of Section 3 has precisely been to introduce the basic ideas and concepts, and to familiarize the reader with the general structure of the algebraic construction before eventually proceeding with more technical tools (Section 4) and finally presenting explicit results (Section 5). Perhaps the presentation flow was not optimal, but all the key ideas were already there in the previous version. In an attempt to improve readability, in the present version we completely reorganized Section 3 and added many additional remarks to make the global picture clearer. We also included the introductory paragraph to Section 3 by briefly explaining the core idea of splitting the problem into the bulk (kinematic) part and the boundary matching conditions. Since there is no well-defined causal order in which the concepts enter the story, we hope that the reader will find this version easier to digest.

On the graded vs. non-graded issue: In our opinion, the readers should appreciate the fact that the so-called fundamental integrable quantum chains with rational R-matrices nicely fit in a common group-theoretic framework. This not only calls for a unified treatment but also conforms with the general philosophy of this work. On the other hand, we cannot recognize any practical value for the splitting suggested by one of our referees: the non-graded cases represent a subclass of solutions which can readily be obtained by simply disregarding the Z_2 phase factor. The auxiliary spaces then become, as explained in the manuscript, those of non-compact sl(2) spins and canonical bosons. The two simplest instances for the non-graded algebras of rank one and two have been explicitly treated in section 5.1.

If our understanding holds, the sole logic behind the splitting is based on the objection by one of the referees claiming that the results for the non-graded su(n)-symmetric quantum chains have been derived in the previous literature and should be separately reviewed. This must be a misconception and we hope that a few remarks below can clarify this. First off, previous works [73,74] (and our subsequent paper [81]) only deal with the solution to the simplest su(2) case (the anisotropic version and slightly more general dissipative boundaries to be more precise). The former two papers, however, aside from having a different scope, do not recognize integrability or make direct use of quantum symmetries. This has been accomplished later in [81]. While the latter is a precursor to our work, extending the construction to higher-rank models was not completely obvious. In fact, in [82] the authors obtained the solution to the su(3) case, but once again have not been able to explain its curious structure on the basis of the underlying quantum symmetry. We nevertheless sincerely believe these previous studies have been acknowledged at appropriate places in the manuscript.

Our manuscript now offers a universal symmetric formulation for a general solution for arbitrary higher-rank algebra, fermionizes the construction and discusses representation-theoretic aspects and the notion of vacuum Q-operators. All of these all original contributions of our work. We believe that this has been effectively communicated to the reader in the abstract and in the final part of the introduction.

2- After carefully re-reading the (previous version of the) manuscript, we have not been able to find any missing 'operative definitions'. We have nevertheless decided to make several extra clarifications regarding the definition of the auxiliary vacua and their internal structure, while also explaining the subtle role of infinite dimensional sl(2) module of highest and lowest weight type. In particular, we now stress multiple times (twice in section 5.1., and in section 5.2.) that the class of solutions under scrutiny always comes with the auxiliary vacuum states which are product states of highest or lowest weight vectors in each irreducible auxiliary component which constitutes the auxiliary algebra of the Lax operator. This in particular means that the solution at hand is uniquely determined after supplying (in addition to Dynkin labels) a set of binary labels which assign the correct types of representations to each irreducible component. As said in our previous reply, this unambiguously defines the vacuum state. Of course, one has to be careful as the very notion of highest- and lowest-weight states is a matter of convention as it suffers from the ambiguities of redefining the generators using algebra automorphisms. For this reason, we give explicit definitions for Verma modules and canonical Fock spaces.

3- We hope that the revised version of the manuscript is typo-free.

---

## Round 4 · List of Changes

1- Slightly revised Introduction.
2- Restructured and refined Section 3.
3- Extra clarifications on the auxiliary vacuum states (Section 5).
4- Removed typos.

---

## Editorial Decision

published